# S$^2$Transformer: Scalable Structured Transformers for Global Station Weather Forecasting

**Hongyi Chen**                                                          *chenhongyi@stu.hit.edu.cn*
*Harbin Institute of Technology, Shenzhen, China*

**Xiucheng Li**$^*$                                                       *lixiucheng@hit.edu.cn*
*Harbin Institute of Technology, Shenzhen, China*

**Xinyang Chen**                                                          *chenxinyang@hit.edu.cn*
*Harbin Institute of Technology, Shenzhen, China*

**Yun Cheng**                                                             *yun.cheng@sdsc.ethz.ch*
*Swiss Data Science Center, Zurich, Switzerland*

**Jing Li**                                                               *li.jing@hit.edu.cn*
*Harbin Institute of Technology, Shenzhen, China*

**Kehai Chen**                                                           *chenkehai@hit.edu.cn*
*Harbin Institute of Technology, Shenzhen, China*

**Liqiang Nie**                                                          *nieliqiang@gmail.com*
*Harbin Institute of Technology, Shenzhen, China*

**Reviewed on OpenReview:** *https://openreview.net/forum?id=AL2VnKno5n*

## Abstract

Global Station Weather Forecasting (GSWF) is a key meteorological research area, critical to energy, aviation, and agriculture. Existing time series forecasting methods often ignore or unidirectionally model spatial correlation when conducting large-scale global station forecasting. This contradicts the intrinsic nature underlying observations of the global weather system, limiting forecast performance. To address this, we propose a novel Spatial Structured Attention Block in this paper. It partitions the spatial graph into a set of subgraphs and instantiates Intra-subgraph Attention to learn local spatial correlation within each subgraph, and aggregates nodes into subgraph representations for message passing among the subgraphs via Inter-subgraph Attention—considering both spatial proximity and global correlation. Building on this block, we develop a multiscale spatiotemporal forecasting model S$^2$Transformer by progressively expanding subgraph scales. The resulting model is both scalable and able to produce structured spatial correlation, and meanwhile, it is easy to implement. The experimental results show that it can achieve performance improvements up to 16.8% over time series forecasting baselines at low running costs. The code is available at `https://github.com/hongyichenhitsz/S2Transformer`.

## 1 Introduction

Global Station Weather Forecasting (GSWF) is vital to modern society, with significant implications for various sectors, including energy (Dehalwar et al., 2016), aviation (Gultepe et al., 2019), and agriculture (Ukhurebor et al., 2022). Unlike regular, image-like data structures such as the Earth Reanalysis 5, weather

---

$^*$Corresponding author.

station data comprises precise, fine-grained meteorological observations distributed across irregular spatial locations. Compared to radars and satellites, weather stations offer greater flexibility in acquiring scattered data and involve lower deployment costs (Wu et al., 2023). Currently, physics-based Numerical Weather Prediction (NWP) models are the most widely used and reputedly the most accurate for GSWF. However, they are inherently computationally intensive, requiring vast resources during execution. Recently, Time Series Forecasting (TSF) methods, which leverage historical weather observations to predict future conditions, have demonstrated superior performance on small-scale weather station datasets and provided a more cost-effective alternative to traditional physics-based prediction models (Han et al., 2024c). However, most of these methods treat GSWF as a multivariate time series forecasting task, trained and tested solely on single-station datasets. They overlook spatial correlation, contradict the intrinsic nature of the global weather system underlying observations, and thus limit forecast performance.

A more appropriate approach is to treat GSWF as a spatiotemporal series forecasting task. Spatiotemporal data is a particular type of multivariate time series data in which each variate is equipped with a spatial coordinate in a metric space. Aside from the general topology shared by multivariate time series, spatiotemporal data also possess the spatial proximity induced by the spatial coordinates, which serves as a strong prior and has a prominent impact on the topology generation. This is summarized by Tobler's first law of geography—everything is related to everything else, but near things are more related than distant things (Miller, 2004). The spatial proximity can be described by a spatial graph, in which each variate is a node, and an edge is drawn between two nodes if their spatial distance is less than a predefined threshold. In the spirit of Tobler's first law, Airphynet (Hettige et al., 2024) and Air-DualODE (Tian et al., 2025) encode the spatial prior into the model by running graph convolution on the spatial graph to make the prediction. Despite the achievement, these methods are mostly trained and tested on meteorological station datasets with localized regions, which limits their applicability in global scenarios. More importantly, it makes distant node pairs very difficult to exchange information due to the limitations (such as over-smoothing and over-squashing) of the message-passing paradigm (Ma et al., 2023). However, Tobler's second law of geography points out that the phenomenon external to a geographic area of interest affects what goes on inside (Tobler, 2004), and many works (Cirstea et al., 2022; Guo et al., 2022) have also confirmed that the distant nodes could also manifest strong correlation. To sidestep this, Corrformer (Wu et al., 2023) organizes global stations into a hierarchical tree structure, with stations under each intermediate node undergoing sequential spatial correlation modeling. Despite achieving linear complexity in modeling global spatial correlation, its unidirectional sequential approach struggles to accommodate the prevalent bidirectional or multidirectional spatial correlation in reality. Inspired by Transformer's milestones in NLP and computer vision, many spatiotemporal forecasting proposals (Bai et al., 2020; Wu et al., 2020; Shang et al., 2021) simply discard the spatial graph and learn spatial correlation end-to-end via an attention mechanism. The attention mechanism enables instant communication between any two nodes, effectively resolving issues with distant message passing and demonstrating significant potential in GSWF. However, such attention-based methods have two drawbacks: 1) The spatial correlation learning is unstructured (lacking structural information embedded in spatial graph) and contains numerous trivial nonzero entries (noise), the accumulated noise is likely to impair the forecasting performance when station number $N$ is large (Appendix C for a formal definition); 2) The computational and memory complexity of generating pairwise spatial correlation both reach $\mathcal{O}(N^2)$, leading to enormous computation costs for large-scale stations.

To address the two limitations, we propose a novel Spatial Structured Attention Block that respects the first law of geography while also permitting long-distance message passing in this paper. The high-level idea is to partition the spatial graph into a set of subgraphs and instantiate self-attention to learn local spatial correlation within each subgraph (Intra-subgraph Attention). To capture the global spatial correlation, we further aggregate the nodes to produce subgraph representations and exchange information among the subgraphs via self-attention again (Inter-subgraph Attention). Since the entire block is differentiable, the message passed from distant nodes can be backpropagated through Inter-subgraph Attention to their correlated nodes in a parsimonious manner. To further enable the perception of spatial structure, we encode the shortest path distance between any two nodes as a bias term in the spatial attention mechanism. Moreover, we stack the proposed Spatial Structured Attention blocks with residual and gradually increase the subgraph scales to develop our eventual GSWF model $S^2$Transformer. Such a design brings the following two appealing features: 1) our proposed method adopts the spatial graph to facilitate the perception and exchange of local

spatial information, and thus it can yield sparse structure and reduce both the computational and memory burdens; 2) it permits message passing between distant node pairs in a parsimonious way, and thus it is capable of capturing global spatial correlation without incurring extra noise. To summarize:

- We propose a novel Spatial Structured Attention Block for GSWF, which not only perceives spatial structure but also considers both spatial proximity and global correlation.

- Building on the proposed block, we develop a multiscale GSWF model $S^2$Transformer by gradually increasing the subgraph scales. The resulting model is scalable and can produce structured spatial correlation.

- Our proposed method is effective yet easy to implement. We evaluate its efficacy and efficiency on global station weather datasets from medium to large sizes. It can achieve performance improvements up to 16.8% over time series forecasting baselines while maintaining low running costs.

## 2 Related Work

### 2.1 Spatiotemporal Series Forecasting

Spatiotemporal series forecasting, a subfield of multivariate time series analysis, has been explored for decades. Common methods for capturing temporal dependencies include recurrent neural networks (RNNs) (Zhao et al., 2017; Lai et al., 2018), convolutional neural networks (CNNs) (Bai et al., 2018; Wu et al., 2022), and Transformer-based models (Vaswani, 2017; Wu et al., 2021; Zhou et al., 2022; Nie et al., 2023). Additionally, multi-layer perceptrons (MLPs) have been applied for time series forecasting (Zeng et al., 2023; Challu et al., 2023; Wang et al., 2024b), showing that even simple models can effectively extract strong temporal periodic patterns. Beyond temporal dependencies, spatial correlation is equally critical in spatiotemporal forecasting. The advancement of graph neural networks (GNNs) offers an effective way to model unstructured spatial adjacency correlation. In the spirit of Tobler's first law of geography, DCRNN (Li et al., 2017) and TGCN (Zhao et al., 2019) leverage a spatial graph based on real-world distance and propose to fuse the local spatial information via graph convolution operation. However, it makes the distant node pairs very hard to exchange information due to the limitations of the message-passing paradigm (Ma et al., 2023), violating Tobler's second law of geography. Subsequently, adaptive GNN-based methods have been proposed to solve this problem. AGCRN (Bai et al., 2020) and MTGNN (Wu et al., 2020) learn a representation for each series and then generate the correlation graph via pairwise node interactions. GTS (Shang et al., 2021) and STEP (Shao et al., 2022b) directly learn a discrete graph based on historical time series. Benefiting from naturally constructing a fully connected graph with learnable edge weights, the self-attention mechanism is also a commonly adopted method for capturing global and dynamic spatial correlation (Jiang et al., 2023; Liu et al., 2024b; Wang et al., 2024c). Nevertheless, the learned spatial correlation matrix in such methods is unstructured (lacking structural information embedded in the spatial graph) and contains a large fraction of trivial nonzero entries (noise). The accumulated noise is likely to impair the forecasting performance when $N$ is large. Moreover, they require $\mathcal{O}(N^2)$ computational complexity, impeding their application in large-scale datasets. Several researchers have developed scalable spatiotemporal forecasting methods to accommodate larger datasets. Detailed related work on this aspect is provided in Appendix A.

### 2.2 Data-driven Numerical Weather Prediction

In recent years, data-driven Numerical Weather Prediction (NWP) models based on machine learning have developed rapidly. Models including Pangu-Weather (Bi et al., 2023), GraphCast (Lam et al., 2023), and Aurora (Bodnar et al., 2025) have demonstrated the ability to surpass conventional physics-based NWP models in terms of forecast accuracy and operational effectiveness. However, as they operate on grid spaces, they may not be optimal for global station weather forecasting. A direct method is to treat GSWF as an independent time series forecasting task and predict meteorological factors for each station individually (Karevan & Suykens, 2020; Hewage et al., 2020; Wu et al., 2021). However, global weather constitutes an integrated system with multi-scale interactions. These methods overlook spatial correlation, contradict the intrinsic nature of the global weather system underlying observations, and thus limit forecast performance.

To address this issue, Airphynet (Hettige et al., 2024) and Air-DualODE (Tian et al., 2025) encode the prior into the model by running graph convolution on the pre-defined spatial graph. Nevertheless, these methods are mostly trained and tested on meteorological station datasets with localized regions. Learning from localized regional data often fails to capture broader spatial patterns. Furthermore, models overfitted to specific regions tend to lack generalization capability, limiting their applicability in real-world scenarios. To accommodate global station weather data, Corrformer (Wu et al., 2023) organizes global stations into a hierarchical tree structure, with stations under each intermediate node undergoing sequential spatial correlation modeling. Despite achieving linear complexity in modeling global spatial correlation, its unidirectional sequential approach struggles to accommodate the prevalent bidirectional or multidirectional spatial correlation in reality.

## 3 Preliminaries

Global station weather data refers to multivariate time series data in which each series is associated with a station's spatial coordinates. Specifically, we define the spatial coordinates of all stations as $\boldsymbol{\lambda} \in \mathbb{R}^N$, $\boldsymbol{\phi} \in \mathbb{R}^N$ and the multivariate time series $\mathbf{X} \in \mathbb{R}^{N \times T \times C}$ records $C$-dimensional meteorological variables collected by $N$ weather stations over $T$ time steps. With a given threshold $\epsilon$, the spatial coordinates of these stations induce a spatial graph $G = (V, E)$, where each node corresponds to a weather station (i.e., $|V| = N$) and two nodes are connected by an edge $e \in E$ if their spatial distance is smaller than $\epsilon$. Besides, we use $\mathbf{A} \in \mathbb{R}^{N \times N}$ to represent the adjacent matrix of $G$. Given the past $T$ steps historical observations $\mathbf{X}_{t-T+1:t}$, along with spatial coordinates $\boldsymbol{\lambda}$, $\boldsymbol{\phi}$ and the spatial adjacent matrix $\mathbf{A}$, the goal of global station weather forecasting is to predict the future $F$ steps of meteorological variables $\hat{\mathbf{X}}_{t+1:t+F}$,

$$\hat{\mathbf{X}}_{t+1:t+F} = \mathcal{F}_\theta(\mathbf{X}_{t-T+1:t}, \boldsymbol{\lambda}, \boldsymbol{\phi}, \mathbf{A}) \tag{1}$$

where $\mathcal{F}_\theta(\cdot)$ denotes the data-driven weather forecasting model parameterized by $\theta$.

## 4 Methodology

The framework of our proposed forecasting model, dubbed Scalable Structured Transformer (S$^2$Transformer), is shown on the left of Figure 1. We first perform spatial graph partitioning for global meteorological stations and calculate their spherical harmonics to extract spatial structure information (Section 4.1). Following the setting of previous work (Liu et al., 2024b; Fang et al., 2025), we reshape and transform the historical observation $\mathbf{X}_{t-T+1:t} \in \mathbb{R}^{N \times T \times C}$ into the input embeddings $\mathbf{X}_{\text{emb}} \in \mathbb{R}^{N \times D}$ with a linear map, where $D$ is the embedding dimension. Next, the embedding and graph partition are fed into the Spatial Structured Attention Block (as described in the right of Figure 1 and Section 4.2) to fuse the information and form the contextual representations, which will be used to make the forecasting. Building on the proposed blocks, we further develop a multi-scale attention mechanism to enhance representation learning by stacking multiple Spatial Structured Attention blocks with various spatial scales (Section 4.3).

### 4.1 Spatial Information Preprocess

In the preprocessing stage, we focus on extracting spatial structure information from input metadata, including spatial graph partitioning and spherical harmonics calculation. Notably, this stage does not introduce scalability bottlenecks. Compared to the training duration, the time spent on preprocessing is negligible. Detailed efficiency analysis is provided in Section 5.3.

**Graph Partition**. Our key observation is that global meteorological observations often exhibit stronger correlation within local areas, which is also supported by Tobler's first law of geography. This motivates us to partition the spatial graph $G$ into a set of subgraphs $\mathcal{G} = \{G_i\}_{i=1}^{P}$, where $G_i = \{V_i, E_i\}$ represents a subgraph of $G$, satisfying $\bigcup G_i = G$ and $\bigcap G_i = \varnothing$. Then, we can fuse the information locally in each subgraph with self-attention. For graph partitioning algorithms, we opt for the METIS algorithm (Karypis & Kumar, 1998) given its efficiency and balanced subgraph outputs.

**Location Embedding**. In addition, location information serves as valuable metadata in numerous geospatial applications, including GSWF. While sine-cosine embedding methods have proven effective in trans-

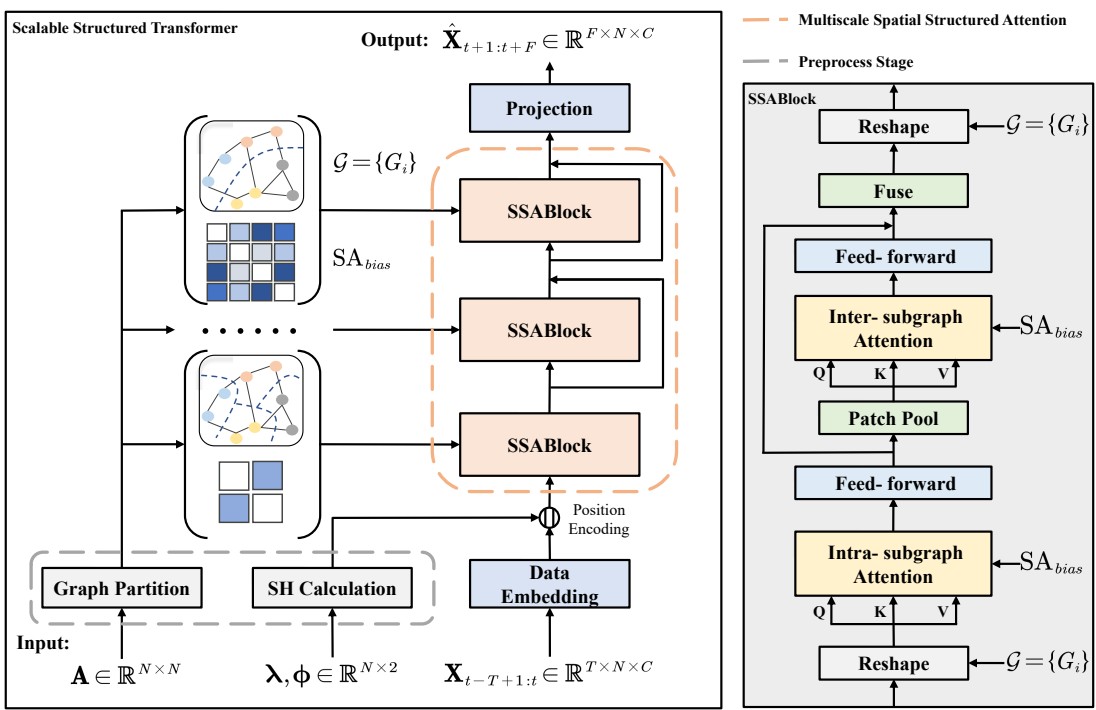

Figure 1: Left: The framework of our proposed Scalable Structured Transformer. Right: The structure of the proposed Spatial Structured Attention Block.

formers, they assume a rectangular domain for longitude and latitude coordinates, which fails to capture the Earth's spherical geometry accurately. Inspired by Geographic Location Encoder (Rußwurm et al., 2024), we employ spherical harmonic basis functions as positional embeddings, which are well-defined globally (including the poles) and enable better discrimination of weather stations distributed across the globe. These spherical harmonics are precomputed from coordinates, with their weights learned directly. Specifically, we define all station location embedding as:

$$\text{SH}(\boldsymbol{\lambda}, \boldsymbol{\phi}) = \oplus_{n=0}^{N} \|_{l=0}^{\infty} \|_{m=-l}^{l} w_l^m Y_l^m(\lambda_n, \phi_n) \tag{2}$$

$$\mathbf{X}_{\text{emb}} = \mathbf{X}_{\text{emb}} \| \text{SH}(\boldsymbol{\lambda}, \boldsymbol{\phi}) \tag{3}$$

where $\oplus$ indicates the stack operator and $\|$ indicates the concatenation operator. $w_l^m$ is a learnable weight shared across all stations and $Y_l^m$ is an orthogonal spherical harmonic basis function with increasingly higher-frequency degrees $l$ and orders $m$. In practice, we choose a maximum number $L$ instead of $\infty$ in Eq. 2. A detailed introduction to spherical harmonics is provided in Appendix B.

## 4.2 Spatial Structured Attention Block

In this section, we introduce details of the Spatial Structured Attention (SSA) Block, guided by Tobler's first and second laws of geography. Firstly, we reshape the input $\mathbf{X}_{\text{emb}}$ from $\mathbb{R}^{N \times D}$ to $\mathbb{R}^{P \times M \times D}$ according to the graph partition result $\mathcal{G}$, where $P$ is the number of subgraphs, and $M$ is the number of nodes in the largest subgraph. If the number of nodes in the subgraph is less than $M$, we pad it with zeros and mask it in subsequent attention operations.

**Intra-subgraph Attention**. In light of Tobler's first law of geography (Miller, 2004), we first employ intra-subgraph attention to model the local spatial correlation. Formally, let $\mathbf{X}$ be the input of the block (e.g., $\mathbf{X}_{\text{emb}}$) and $\mathbf{X}_p \in \mathbb{R}^{M \times D}$ denote the embedding in the $p$-th subgraph, the representations within the

subgraph are updated as follows.

$$\boldsymbol{\alpha}_p = \text{softmax}\left(\frac{\mathbf{Q}_p \mathbf{K}_p^\top}{\sqrt{d}}\right) \tag{4}$$

$$\mathbf{Y}_p = \text{FFN}\left(\boldsymbol{\alpha}_p \mathbf{V}_p\right) \tag{5}$$

$$\mathbf{Q}_p, \mathbf{K}_p, \mathbf{V}_p = (\mathbf{W}_Q, \mathbf{W}_K, \mathbf{W}_V)\mathbf{X}_p \tag{6}$$

where $\mathbf{W}_Q$, $\mathbf{W}_K$ and $\mathbf{W}_V$ are learnable parameters that transform $\mathbf{X}_p$ into different semantic spaces, $\boldsymbol{\alpha}_p \in \mathbb{R}^{M \times M}$ is the intra-subgraph attention map that captures the local spatial correlation in the $p$-th subgraph, and $\mathbf{Y}_p$ is the learned contextual representations that will be fed to the subsequent inter-subgraph attention.

**Inter-subgraph Attention**. Tobler's second law of geography (Tobler, 2004) states that the phenomenon external to a geographic area of interest affects what goes on inside. However, as mentioned in Section 1, directly using the attention mechanism will lead to quadratic complexity $\mathcal{O}(N^2)$ and incur extra noise that impairs the forecasting performance. Thus, we innovatively learn the attention between different subgraphs to approximate the global attention mechanism. We first apply mean pooling to $\mathbf{Y}_p$ to obtain the subgraph representation $\mathbf{s}_p \in \mathbb{R}^D$ and then stack them to produce $\mathbf{S} \triangleq [\mathbf{s}_1; \mathbf{s}_2; \dots; \mathbf{s}_P] \in \mathbb{R}^{P \times D}$. Then, we employ the inter-subgraph attention to exchange information across subgraphs as follows.

$$\boldsymbol{\alpha}' = \text{softmax}\left(\frac{\mathbf{Q}'\mathbf{K}'^\top}{\sqrt{d}}\right) \tag{7}$$

$$\mathbf{S}' = \text{FFN}\left(\boldsymbol{\alpha}'\mathbf{V}'\right) \tag{8}$$

$$\mathbf{Q}', \mathbf{K}', \mathbf{V}' = (\mathbf{W}_{Q'}, \mathbf{W}_{K'}, \mathbf{W}_{V'})\mathbf{S} \tag{9}$$

where $\mathbf{W}_{Q'}$, $\mathbf{W}_{K'}$, and $\mathbf{W}_{V'}$ are learnable parameters. $\boldsymbol{\alpha}' \in \mathbb{R}^{P \times P}$ captures the global spatial correlation among subgraphs, which allows us to approximate the global attention mechanism by assigning the same attention value $\alpha_{pq}$ to any pair of nodes from the subgraphs $p$ and $q$. This can be considered as a sort of *implicit regularization* that encourages the model to focus on mining groups of nodes that share similar patterns while ignoring unrelated noise. In other words, we only permit the distant node pairs to exchange information in a parsimonious manner, and only valuable information can be transmitted across groups.

In the end, to take both local and global spatial information into account, we expand the shape of $\mathbf{S}'$ in Eq. 8 from $\mathbb{R}^{P \times D}$ to $\mathbb{R}^{P \times M \times D}$, which is concatenated with the local representation and then transformed by a linear map to produce the output:

$$\mathbf{X}' = \mathbf{W}(\mathbf{Y}\|\mathbf{S}') \tag{10}$$

where $\|$ indicates the concatenation operator and $\mathbf{W} \in \mathbb{R}^{2D \times D}$ is the parameter of linear map layer.

**Spatial Attention Bias**. Self-attention serves as the core computational module in our proposed block, however, it is oblivious to the local graph structures due to its permutation invariant property. To encode the structural information into the attention mechanism, we encode the shortest path distance between any two nodes as a bias in spatial attention inspired by Graphormer (Ying et al., 2021). Specifically, we precompute the shortest path distance (SPD) between two stations. For unconnected stations, the SPD is set to a special value of $-1$. We calculate the spatial attention bias matrix $\mathbf{SA}^{\mathbf{bias}}$ by element-wise embedding the SPD matrix with a learnable scalar. We then revise Eq. 4 as follows:

$$\boldsymbol{\alpha}_p = \text{softmax}\left(\frac{\mathbf{Q}_p \mathbf{K}_p^\top}{\sqrt{d}} + \mathbf{SA}_p^{\text{bias}}\right) \tag{11}$$

$$\mathbf{SA}_p^{\text{bias}} = \sigma\left(\text{SPD}\left(\mathbf{A}_p\right)\right) \tag{12}$$

where $\mathbf{SA}_p^{\text{bias}} \in \mathbb{R}^{M \times M}$ is the intra-subgraph attention bias matrix in the $p$-th subgraph and $\sigma$ is an element-wise learnable scalar shared across all blocks. We can modify Eq. 7 in the same manner.

Table 1: Dataset statistics.

| Dataset | Frequency | Time Span | Stations | Variables Name |
|---------|-----------|-----------|----------|----------------|
| WEATHER-5K | 1 hour | 2014-2023 | 5672 | Temp, Dewpoint, Wind Rate, Wind Direc, Sea Level |
| NCEI Global | 1 hour | 2019-2020 | 3850 | Wind, Temp |

### 4.3 Multiscale Spatial Structured Architecture

In practice, the spatial correlation often presents multiscale structures due to the multiscale property of underlying physical dynamics. To capture the intrinsic multiscale property, we develop a multiscale spatial balance architecture by stacking $L$ SSA blocks by gradually increasing the subgraph scales. Specifically, we perform graph partition with various subgraph scales to produce $L$ sets, $\mathcal{G}_1, \mathcal{G}_2, \ldots, \mathcal{G}_L$ such that $|\mathcal{G}_i| = |\mathcal{G}_{i-1}|/2$. $L$ is set to 2 by default, and we empirically find that it performs quite well in practice. Such a design also brings two additional benefits: 1) it progressively expands the receptive field of the node attention mechanism, which aligns with the spatial diffusion process of the global weather system's physical dynamics; 2) it offers the chance for spatially closed nodes at the boundary of two subgraphs to exchange information in the high-level block.

**Forecasting**. We reshape the output of $L$-th block $\mathbf{X}'^{(L)}$ from $\mathbb{R}^{P \times M \times D}$ to $\mathbb{R}^{N \times D}$ according to the graph partition result and produce the multi-step prediction $\hat{\mathbf{X}}_{t_0:t_0+F}$ through a linear projection. The model is optimized by minimizing the mean absolute error:

$$\mathcal{L}(\mathbf{X}_{t_0:t_0+F}, \hat{\mathbf{X}}_{t_0:t_0+F}) = \frac{\sum_{n=1}^{N} \sum_{t=t_0}^{t_0+F-1} \sum_{c=0}^{C} |\hat{x}_{n,t,c} - x_{n,t,c}|}{N \times F \times C} \tag{13}$$

**Complexity Analysis**. Our method achieves significant computational efficiency improvements by focusing on Intra-subgraph and Inter-subgraph Attention. Intra-subgraph Attention (for subgraphs with $\frac{N}{P}$ nodes) has complexity $\mathcal{O}(P(\frac{N}{P})^2 D) = \mathcal{O}(\frac{N^2}{P}D)$, and Inter-subgraph Attention (for cross-subgraph correlation) has $\mathcal{O}(P^2 D)$, leading to an overall complexity of $\mathcal{O}(\frac{N^2}{P}D + P^2 D)$. Minimizing this (via derivative calculation) gives $\mathcal{O}(2N^{4/3}D)$ when $P = N^{2/3}$, with better scalability than quadratic-complexity methods. Our model's efficiency can be further enhanced by linear attention; though Corrformer (specifically designed for GSWF tasks) has a better theoretical complexity of $\mathcal{O}(NT \log TD)$, it performs poorly in practice. This is due to its temporal alignment/rearrangement operations (for inferred node order), which disrupt data layout, increase memory latency (hindering hardware parallelism), and raise memory occupancy (via intermediate data storage). Detailed efficiency analysis is provided in Section 5.3.

## 5 Experiments

In this section, we evaluate our approach using two benchmark datasets (Section 5.2 and Appendix E). Section 5.3 presents the efficiency analysis, and Section 5.4 describes the ablation study. A more comprehensive analysis of the graph partitioning strategy is provided in Appendix F. The sensitivity of hyperparameters is detailed in Section 5.5 and Appendix G. To gain a more profound understanding of our model, we also conducted visualizations, which are included in Section 5.6 and Appendix H.

### 5.1 Experimental Setup

**Datasets**. We evaluate the performance and efficiency of the proposed method on two global station weather forecasting benchmarks. The first benchmark is the WEATHER-5K dataset (Han et al., 2024c), which includes crucial weather elements collected from 5672 global weather stations over ten years. The second benchmark is the NCEI Global dataset (Wu et al., 2023), which contains the hourly averaged wind speed and hourly temperature of 3,850 stations worldwide from 2019 to 2020. Dataset statistics are presented in Table 1, and further particulars are available in Appendix D.

Table 2: Global station weather forecasting performance comparison. The best results are highlighted in **bold**, while the second-best results are underlined.

| Dataset | WEATHER-5K | | | | | | | | | | NCEI Global | | | |
|---|---|---|---|---|---|---|---|---|---|---|---|---|---|---|
| Variable | Temperature | | Dewpoint | | Wind Rate | | Wind Direc. | | Sea Level | | Wind | | Temp | |
| Metric | MAE | MSE | MAE | MSE | MAE | MSE | MAE | MSE | MAE | MSE | MAE | MSE | MAE | MSE |
| NWP Model | 1.76 | 7.39 | 1.85 | 7.94 | 1.48 | 4.53 | 63.8 | 7158.3 | **0.86** | **2.68** | 1.59 | 5.00 | 1.91 | 13.45 |
| Informer | 1.88 | 7.51 | 1.94 | 8.30 | 1.30 | 3.62 | 60.7 | 6906.9 | 2.01 | 10.56 | 1.58 | 4.93 | 4.42 | 33.29 |
| Autoformer | 1.93 | 8.64 | 2.06 | 9.57 | 1.42 | 3.97 | 66.5 | 7710.0 | 2.26 | 12.78 | 1.47 | 4.69 | 2.25 | 10.14 |
| Pyraformer | 1.75 | 6.92 | 1.83 | 7.88 | 1.30 | 3.58 | 61.8 | 6930.2 | 1.90 | 9.72 | 1.51 | 4.61 | 3.67 | 23.33 |
| STID | 1.78 | 7.09 | 1.83 | 7.88 | 1.28 | 3.53 | 60.9 | 6722.2 | 1.87 | 9.42 | 1.34 | 3.83 | 1.99 | 8.46 |
| MTGNN | 1.84 | 7.36 | 1.89 | 8.18 | 1.30 | 3.59 | 62.1 | 6854.5 | 1.91 | 9.64 | 1.37 | 3.90 | 2.07 | 8.51 |
| Corrformer | 1.99 | 8.21 | 2.09 | 9.47 | 1.38 | 3.83 | 66.7 | 7832.3 | 2.19 | 12.39 | **1.30** | 3.89 | 1.89 | 7.71 |
| iTransformer | 1.64 | 5.94 | 1.67 | 6.57 | 1.24 | 3.31 | 58.6 | 6570.6 | 1.47 | 5.53 | 1.33 | 3.87 | 1.90 | **7.50** |
| RPMixer | 1.77 | 6.60 | 1.83 | 7.45 | 1.28 | 3.52 | 60.3 | 6607.1 | 1.65 | 6.45 | 1.43 | 4.02 | 2.47 | 11.08 |
| PatchSTG | 1.65 | 5.94 | 1.68 | 6.58 | 1.20 | 3.15 | 57.2 | 6254.8 | 1.41 | 4.92 | 1.36 | 3.89 | 2.21 | 9.59 |
| S²Transformer | **1.47** | **4.99** | **1.52** | **5.67** | **1.17** | **3.09** | **55.5** | **6135.7** | 1.26 | 4.08 | **1.30** | **3.61** | **1.87** | **7.50** |

**Baselines**. We compare our method with the following baselines: (1) Physics-based NWP model: ECMWF-HRES (EC) for WEATHER-5K dataset and ERA5 (reanalysis, 0.25°) (Hersbach et al., 2020) for NCEI Global dataset; (2) Pure time dependencies modeling methods: Informer (Zhou et al., 2021), Autoformer (Wu et al., 2021), Pyraformer (Liu et al., 2022), STID (Shao et al., 2022a); (3) Spatial correlation modeling methods: MTGNN (Wu et al., 2020), Corrformer (Wu et al., 2023), iTransformer (Liu et al., 2024b); (4) Scalable spatial correlation modeling methods: RPMixer Yeh et al. (2024), PatchSTG (Fang et al., 2025). Notably, Pyraformer and Corrformer are the Best Time Series Forecasting methods reported on the WEATHER-5K and NCEI Global datasets. More details of baselines are provided in Appendix D.

**Evaluation Metrics**. We conduct a comprehensive comparison using various evaluation criteria from the performance and efficiency perspectives. We evaluated performance using the mean absolute error (MAE) and mean square error (MSE). We consider efficiency by measuring both the training wall-clock time and maximum memory usage during training.

**Implementation details**. Given our focus on short-term global station weather forecasting, we predict one day into the future using the past two days of data, where the input length is 48 (hours) and the predicted length is 24 (hours). The key parameter settings are detailed in Appendix D. All experiments in this study are implemented using PyTorch Paszke et al. (2019) and conducted on an NVIDIA RTX 4090 GPU with 24GB memory. We run each experiment three times and report the average results.

## 5.2 Performance Comparison

Table 2 presents the average forecast performance in 24 hours of all methods with an input length of 48 hours. Notably, following previous studies, the models on the WEATHER-5K dataset adopt a unified architecture to predict all variables at once, while those on the NCEI Global dataset are trained and tested separately for each variable. The best results are highlighted in **bold** and the second-best in underlined. To ensure a fair comparison, all baselines are implemented with their official configurations. The experimental conclusions are as follows:

First, although current physics-based NWP models are regarded as the most accurate weather forecasting models, time series forecasting methods have demonstrated comparable performance in short-term forecast-

Table 3: Global station weather forecasting efficiency comparison. BS: batch size used in model training. Mem: max memory used during training (in gigabytes). Time: total training time (in hours). The best results are highlighted in **bold**.

| Dataset | WEATHER-5K | | | NCEI Global | | |
|---|---|---|---|---|---|---|
| Metric | BS | Mem | Time | BS | Mem | Time |
| MTGNN | **64** | 179.3 | 12.7 | **64** | 64.3 | 0.6 |
| Corrformer | 8 | 143.6 | 29.4 | 8 | 143.2 | 9.0 |
| iTransformer | 16 | 151.2 | 7.5 | 24 | 159.5 | 0.3 |
| $S^2$Transformer | **64** | **52.0** | **5.0** | **64** | **43.3** | **0.2** |

ing tasks. Second, among TSF methods, the suboptimal performance of Informer, Autoformer, Pyraformer, and STID highlights the critical role of spatial correlation modeling. In contrast, spatial correlation modeling methods (MTGNN, iTransformer, and Corrformer) perform more effectively on the medium-scale NCEI Global dataset, underscoring the significance of this modeling paradigm. Our method outperforms all baselines, delivering up to a **5.7%** performance improvement on the Wind variable of the NCEI Global dataset, attributed to its structured dynamic spatial correlation modeling. Furthermore, the limitations of the aforementioned models, such as the quadratic complexity of MTGNN and iTransformer, and the large parameter scale of Corrformer due to its complex structure, restrict their application to the large-scale WEATHER-5K dataset, further emphasizing the advantages of scalable spatial correlation modeling approaches. However, methods that simplify spatial receptive fields (PatchSTG) and mix channels in the spatial dimension (RP-Mixer) both suffer from information loss, caused by insensitivity to spatial structural information or spatial downsampling. In contrast, our method achieves state-of-the-art performance across most variables on the WEATHER-5K dataset, with an improvement of up to **16.8%** on the Temperature variable. This superior performance stems from the structured spatial correlation constraints of our method: accurate perception and capture of local spatial correlations and restricted global spatial correlations, which preserve key information and filter out noise. We further provide the results and analysis of long-term global station weather forecasting, along with a comparison with physics-informed models, in Appendix E.

## 5.3 Efficiency Study

First, our proposed model consists of two stages: preprocessing and training. Verified by multiple experiments, the preprocessing time for both datasets is stably within 2 minutes. Compared with the training duration, the preprocessing time is negligible. To further evaluate model efficiency, we compare our method with existing spatial correlation modeling methods in three key metrics: **batch size**, **maximum memory usage during training**, and **total training time consumption**. The models are trained with multi-GPU parallel acceleration using *torch.nn.DataParallel*. As shown in Table 3, the key observations are as follows.

Existing spatial correlation modeling models (MTGNN, iTransformer) exhibit rapid growth in computational and/or memory consumption as the number of nodes increases, primarily due to their quadratic complexity. Although Corrformer only models unidirectional spatial correlations, its complex Encoder-Decoder architecture (different from the aforementioned Encoder-Only models) leads to a large memory footprint, which further results in a smaller batch size and longer training time. In contrast, the proposed model in this paper confines fine-grained spatial correlation modeling within local subgraphs, enabling efficient long-distance information exchange via inter-subgraph spatial correlation modeling. This design not only filters out noise to improve performance but also effectively reduces memory consumption. Across the two datasets, the efficiency ranking of all models remains consistent: $S^2$Transformer > iTransformer > MTGNN > Corrformer. Particularly, compared with Corrformer (specifically designed for GSWF tasks), $S^2$Transformer reduces memory consumption by approximately **64**% and improves inference speed by about **83**% on the WEATHER-5K dataset; on the NCEI Global dataset, it achieves a memory reduction of around **70**% and an inference speedup of roughly **98**%.

Table 4: Ablation study on the WEATHER-5K dataset. The best results are highlighted in **bold**.

| Variable | Temperature | | Dewpoint | | Wind Rate | | Wind Direc. | | Sea Level | |
|---|---|---|---|---|---|---|---|---|---|---|
| Metric | MAE | MSE | MAE | MSE | MAE | MSE | MAE | MSE | MAE | MSE |
| w/o. Metis | 1.64 | 5.99 | 1.71 | 6.83 | 1.25 | 3.40 | 59.25 | 6607.08 | 1.72 | 7.88 |
| w/o. SH | 1.49 | 5.08 | 1.54 | 5.75 | 1.18 | 3.09 | 56.02 | 6206.36 | 1.30 | 4.37 |
| w/o. Intra-Att | 1.65 | 6.03 | 1.70 | 6.77 | 1.26 | 3.44 | 59.74 | 6639.95 | 1.53 | 6.02 |
| w/o. Inter-Att | 1.57 | 5.63 | 1.62 | 6.34 | 1.23 | 3.30 | 58.44 | 6493.64 | 1.52 | 6.30 |
| w/o. SA | 1.48 | 5.03 | 1.54 | 5.72 | 1.18 | **3.06** | 55.75 | 6171.15 | **1.26** | 4.13 |
| w/o. MA | 1.59 | 5.12 | 1.57 | 5.81 | 1.19 | 3.10 | 55.86 | 6188.84 | 1.28 | 4.18 |
| $S^2$Transformer | **1.47** | **4.99** | **1.52** | **5.67** | **1.17** | 3.09 | **55.50** | **6135.69** | **1.26** | **4.08** |

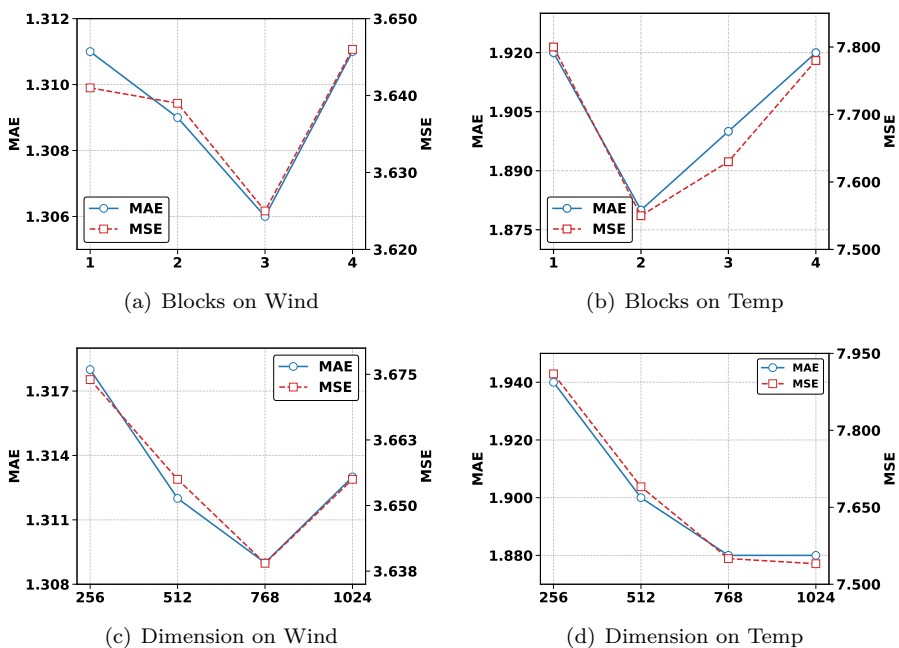

(a) Blocks on Wind

(b) Blocks on Temp

(c) Dimension on Wind

(d) Dimension on Temp

Figure 2: Parameter sensitivity analysis

## 5.4 Ablation Study

We perform an ablation study on the WEATHER-5K dataset to validate the effectiveness of the proposed modules. Specifically, we consider the following variants of our proposed model: (1) **w/o. Metis**: The METIS graph partitioning method is substituted with random partitioning. (2) **w/o. SH**: The spherical harmonic position encoding in the preprocessing stage is removed. (3) **w/o. Intra-Att**: The intra-subgraph attention is removed, and only the spatial correlation between subgraphs is captured. (4) **w/o. Inter-Att**: The inter-subgraph attention is removed, and only the local spatial correlation is captured. (5) **w/o. SA**: The spatial attention bias matrix in both inter-subgraph and intra-subgraph attention is removed. (6) **w/o. MA**: The hierarchical multi-scale architecture is superseded by a sequence of single-scale blocks. As shown in Table 4, the key observations are as follows.

First, the performance drop of **w/o. Metis** confirms the importance of graph partitioning algorithms. METIS, a hierarchical partitioning algorithm, ensures subgraph balance with only the partition count as a parameter. Second, **w/o. Intra-Att** shows the most severe performance decline, indicating that local

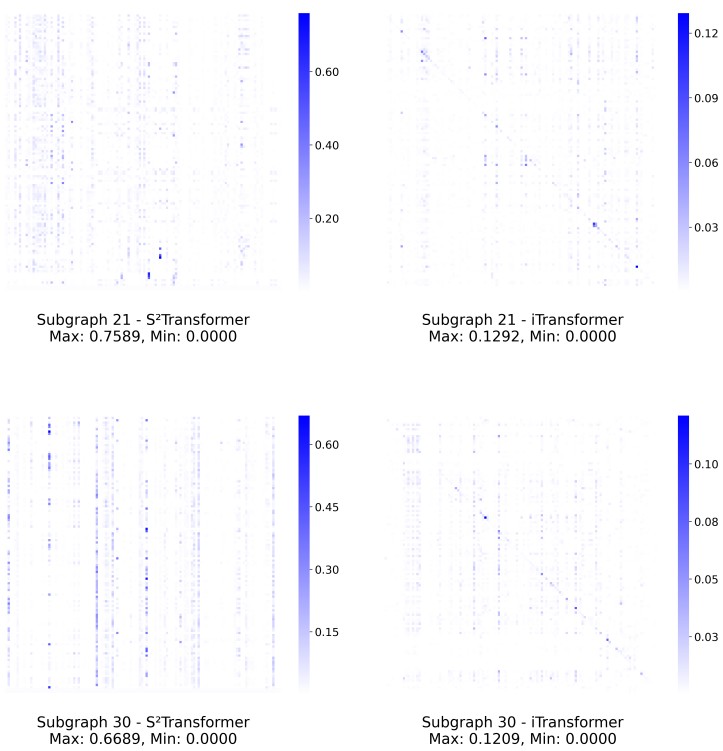

Figure 3: Visualization of intra-subgraph attention matrices for $S^2$Transformer and iTransformer.

spatial correlation is critical for accurate global station weather forecasting, consistent with Tobler's first law of geography. Meanwhile, **w/o. Inter-Att** exhibits the significant drop, suggesting global spatial correlation also aids precise forecasting, aligning with Tobler's second law of geography. Finally, the performance of **w/o. SH**, **w/o. SA** and **w/o. MA** validate the effectiveness of the proposed spherical harmonic position encoding, spatial attention bias, and structured scale progression in capturing spatial correlation.

## 5.5 Parameter Sensitivity Analysis

We evaluate the sensitivity of hyperparameters (including the number of blocks $L$ and embedding dimension $D$) on the NCEI Global dataset. Figure 2 shows that model performance improves with increasing $L$, achieving the best results at $L = 2$ for the Global Temperature dataset and $L = 3$ for the Global Wind dataset. Further increasing $L$ leads to a degradation in performance. Thus, we select $L = 2$ for model efficiency. Similarly, optimal performance is achieved with an embedding dimension of $D = 768$, with no further performance gain from increasing $D$. This indicates that a small model suffices to learn spatiotemporal knowledge in global station weather forecasting. The sensitivity analysis of look-back window length $T$ and the number of subgraphs $P$ is presented in Appendix G.

## 5.6 Visualization

To gain deeper insights into our proposed model, we conducted supplementary visualization experiments, focusing on a comparative analysis of the intra-subgraph attention matrix of our model ($S^2$Transformer) and the attention matrix of the iTransformer model. Specifically, we selected the intra-subgraph attention matrices corresponding to two subgraphs from the test data. For the purpose of comparison, we extracted the relevant subgraph attention matrix fragments from iTransformer's global attention matrix, with the extraction process strictly guided by the node indices of the two aforementioned subgraphs.

As illustrated in Figure 3, owing to the global receptive field of iTransformer's attention mechanism, the attention values in its corresponding subgraph regions are scattered and small (with a maximum of approxi-

mately 0.12). When the number of such scattered trivial values is large, they tend to introduce trivial noise and impair model performance. In contrast, our model proactively restricts the receptive field to confine attention within the subgraph—an approach that not only aligns with relevant geographical laws but also yields more concentrated and larger attention values (with a broader distribution of attention weights). This confirms that our model effectively avoids noise introduction during local modeling. The quantitative discrepancy between this structured spatial correlation and iTransformer's unstructured attention serves as direct evidence of noise reduction, validating the rationality of our model design. We provide additional visualization analysis results in Appendix H.

## 6 Conclusion and Limitation

In this paper, we propose a novel Spatial Structured Attention Block that not only perceives spatial structure but also considers both spatial proximity and global correlation. Specifically, we partition the spatial graph into a set of subgraphs and utilize the Intra-subgraph and Inter-subgraph Attention to learn local and global spatial correlation. Building on the proposed block, we develop a multiscale model $S^2$Transformer by progressively increasing the subgraph scales. The resulting model is scalable and can produce structured spatial correlation. Performance comparison and efficiency analysis validate the superiority of our method in medium and large-scale global station weather forecasting scenarios. Ablation studies confirm the effectiveness of the model designs, and we further explore the hyperparameters in model construction.

**Limitation**. However, we noticed that our model still lags behind the numerical weather prediction model for longer lead times. This is mainly because an increase in forecast duration amplifies the non-linear dynamics of the atmospheric system, and pure data-driven methods struggle to fully capture their evolutionary laws. In the future, we would like to explore the nascent regime of data-driven and physics-informed paradigms to enhance the model's ability to predict long-term weather processes.

### Broader Impact Statement

In this paper, we propose a novel scalable spatiotemporal series forecasting model that captures structured spatial correlation guided by Tobler's laws of geography to enhance global station weather forecasting while maintaining low running costs. By mitigating resource barriers, this framework facilitates the responsible deployment of climate and disaster forecasting in underserved regions. Ultimately, our work democratizes access to reliable predictive tools, fostering global resilience and data-driven disaster management.

### Acknowledgments

This work is supported by Guangdong Basic and Applied Basic Research Foundation under Grant No. 2025A1515012932, Shenzhen College Stability Support Plan under Grant No. GXWD20220811173233001, Shenzhen Science and Technology Program No. JCYJ20241202123503005, ZDSYS20230626091203008, and KQTD20240729102154066.

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

## A Scalable Spatiotemporal Series Forecasting

Several researchers have developed scalable spatiotemporal forecasting methods to accommodate larger datasets. Model-agnostic approaches use graph partitioning to decompose large spatial graphs into smaller subgraphs, with experiments conducted via independent training (Mallick et al., 2020) or continual learning (Wang et al., 2024a). In contrast, existing scalable models fall into four paradigms: precomputing spatial correlation, linearizing spatial correlation computation, mixing channels across spatial dimensions, and simplifying nodes' spatial receptive fields. SGP (Cini et al., 2023) and SimST (Liu et al., 2024a) precompute graph convolutions and decouple spatial correlation modeling from training, but their fixed input-space representations may reduce effectiveness. BigST (Han et al., 2024a) and Sumba (Chen et al., 2024) adopt linearized spatial convolutions to lower complexity, yet low-rank approximations prevent them from capturing structured spatial correlation. The channel mixing approach (Yeh et al., 2024; Han et al., 2024b) enhances scalability by aggregating and routing messages across dimensions, avoiding quadratic complexity but suffering from information dilution during aggregation, leading to suboptimal practical performance. Methods simplifying spatial receptive fields, such as SAGDFN (Jiang et al., 2024), use significant neighbor sampling to model spatial correlation but fail to preserve local structural information. Similarly, PatchSTG (Fang et al., 2025) partitions traffic nodes into non-overlapping KDTree-based patches, using depth attention for local correlation and breadth attention across same-index patches for global aggregation. However, lacking inherent integration of spatial prior knowledge (e.g., topological connections, geographic proximity), it overly relies on training patterns and struggles to capture critical geographic dependencies.

## B Spherical Harmonics

Spherical harmonics have been widely used in Earth science (Klosko & Wagner, 1982; Pail et al., 2011; Thébault et al., 2021). Any function $f(\lambda, \phi)$ defined on a sphere can be expressed as a weighted sum of orthogonal spherical harmonic basis functions $Y_l^m$ with increasing frequency, characterized by degrees $l$ and orders $m$:

$$f(\lambda, \phi) = \sum_{l=0}^{\infty} \sum_{m=-l}^{l} w_l^m Y_l^m(\lambda, \phi) \tag{14}$$

Here, $w_l^m$ are the weights. Each spherical harmonic $Y_l^m$ is defined as:

$$Y_l^m(\lambda, \phi) = \sqrt{\frac{2l+1}{4\pi} \cdot \frac{(l-|m|)!}{(l+|m|)!}} \, P_l^m(\cos \lambda) \, e^{im\phi} \tag{15}$$

where $P_l^m(x)$ denotes the associated Legendre polynomials, given by:

$$P_l^m(x) = (-1)^m (1-x^2)^{\frac{m}{2}} \frac{d^m}{dx^m} P_l(x) \tag{16}$$

These involve derivatives of Legendre polynomials $P_l(x)$, which are defined as:

$$P_l(x) = \frac{1}{2^l l!} \frac{d^l}{dx^l} (x^2 - 1)^l \tag{17}$$

For associated Legendre polynomials of negative order ($m < 0$), the symmetry relation can be used.

$$P_l^{-m}(x) = (-1)^m \frac{(l-m)!}{(l+m)!} P_l^m(x) \tag{18}$$

In practice, we use the real form of Eq. 15:

$$Y_l^m(\lambda, \phi) = \begin{cases} (-1)^m \sqrt{2} \bar{P}_l^{|m|}(\cos \lambda) \sin(|m|\phi), & \text{if } m < 0, \\ \bar{P}_l^m(\cos \lambda), & \text{if } m = 0, \\ (-1)^m \sqrt{2} \bar{P}_l^m(\cos \lambda) \cos(m\phi), & \text{if } m > 0, \end{cases} \tag{19}$$

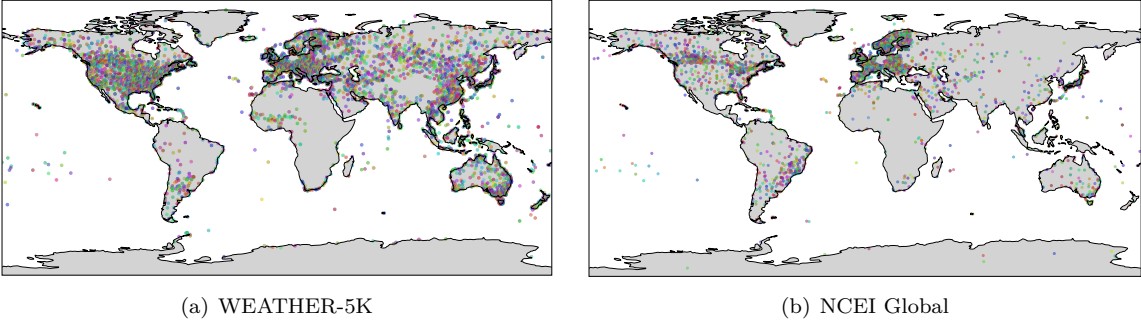

(a) WEATHER-5K

(b) NCEI Global

Figure 4: Global meteorological station distribution

where $\bar{P}_l^m(\cos\lambda)$ denotes the normalized associated Legendre polynomial, defined as:

$$\bar{P}_l^m(\cos\lambda) = \sqrt{\frac{2l+1}{4\pi} \cdot \frac{(l-|m|)!}{(l+|m|)!}} P_l^m(\cos\lambda) \tag{20}$$

## C  Structured Spatial Correlation

In spatiotemporal forecasting tasks, structured spatial correlation refers to a learned dependency pattern that aligns with the inherent physical topology of the network rather than being a collection of arbitrary, dense, or noisy associations. To provide a precise characterization, we define a spatial correlation matrix $\Theta \in \mathbb{R}^{N \times N}$ (where $\Theta_{ij}$ represents the learned correlation strength between nodes $v_i$ and $v_j$) as structured if it satisfies the following two constraints:

- **Geographic Proximity Constraint**: For any station $v_i$, the cumulative correlation strength of its geographic $\epsilon$-neighborhood $\mathcal{N}_\epsilon(v_i)$ accounts for at least a threshold proportion $\tau$ of its total correlation strength:

$$\frac{\sum_{v_j \in \mathcal{N}_\epsilon(v_i)} \Theta_{ij}}{\sum_{v_j \in V} \Theta_{ij}} \geq \tau \tag{21}$$

- **Sparsity Constraint**: The sparsity of $\Theta$ is bounded by a threshold $\rho$ to eliminate trivial noise:

$$\text{Sparsity}(\Theta) = \frac{|\{(i,j) \mid \Theta_{ij} > 0\}|}{N^2} \leq \rho \tag{22}$$

## D  Implementation Details

### D.1  Datasets

All datasets used in our study are open-source or freely available for research purposes.

**WEATHER-5K**. The WEATHER-5K dataset, derived from global near-surface in-situ observations via the public Integrated Surface Database (ISD), includes data from 5,672 high-quality weather stations (2014-2023) covering variables like wind speed, direction, and temperature. It undergoes rigorous quality control (data interpretation, temporal alignment, completeness filtering, outlier detection), with remaining missing data interpolated using ERA5, followed by standardization and extreme value percentile calculation. Featuring uneven station distribution, it better aligns with actual observations than simulated ERA5 data, addressing limitations of small-scale existing time-series meteorological datasets for relevant research. The dataset can be accessed at `https://github.com/taohan10200/WEATHER-5K`.

**NCEI Global**. The NCEI Global dataset, sourced from the National Centers for Environmental Information, comprises hourly averaged wind speed and temperature records from 3,850 stations across the

globe, covering the period from January 1, 2019, to December 31, 2020. It is divided into two subsets: "global wind" and "global temp". The dataset is publicly accessible via the National Oceanic and Atmospheric Administration (NOAA) at `https://www.ncei.noaa.gov/data/global-hourly/access`. We utilize the available processed versions in this study, which are available in the Corrformer GitHub repository: `https://github.com/thuml/Corrformer`.

We visualize the station distributions of both datasets in Figure 4, which shows that the stations effectively cover diverse weather patterns across varying geographical scales and station densities. Following previous research, we chronologically split the WEATHER-5K dataset into training, validation, and test sets at a ratio of 0.8/0.1/0.1, and the NCEI Global dataset at 0.7/0.1/0.2. Input data were normalized using the Z-score for model training.

## D.2 Baselines

We compare the proposed approach with physics-based NWP models and the following advanced time series forecasting baselines:

- Informer: It utilizes ProbSparse self-attention to reduce computational complexity, enabling efficient long-sequence forecasting by focusing on dominant temporal patterns.

- Autoformer: It employs a decomposition-attention architecture to model trend and seasonal components, leveraging an auto-correlation mechanism to capture long-range dependencies without explicit alignment.

- Pyraformer: It uses a hierarchical pyramid graph to capture multi-scale temporal dependencies with linear complexity, enabling efficient long-range forecasting.

- STID: It combines spatial and temporal identity embeddings with multi-layer perceptrons to address sample indistinguishability in spatiotemporal dimensions.

- MTGNN: It builds an adaptive static global directed graph using learnable node embedding and aggregates information along spatial dimensions through mix-hop propagation.

- Corrformer: It integrates multi-correlation mechanisms (spatial cross-correlation and temporal auto-correlation) in a learnable tree structure to model complex spatiotemporal dependencies for large-scale global station weather forecasting.

- iTransformer: It embeds the whole time series into a spatial token and captures dynamic global spatial correlation using the self-attention mechanism.

- RPMixer: It employs MLPs to model temporal dependency and integrates random projection layers to capture spatial correlation.

- PatchSTG: It uses irregular spatial patching via KDTree and dual attention (depth and breadth) to capture local and global spatial correlations in large-scale spatiotemporal networks.

We obtain the code of baselines directly from their corresponding GitHub repositories. For the model- and training-related configurations, we follow the recommended settings provided in their code.

## D.3 Hyperparameters Setting

To better reproduce our model, we summarize all the default hyperparameters as follows. The dimensions of the input embedding and hidden embedding dimension $D$ are set to 768. The number of blocks $L$ is set to 2. The initial number of subgraphs $P$ is set based on hyperparameter tuning results. Specifically, 64 for the WATHER-5K and NCEI Global Wind dataset, and 16 for the NCEI Global Temp dataset. In the calculation of spherical harmonic basis functions, we set the maximum order $l$ of Legendre polynomials to 3. The epsilon $\epsilon$ is set by following the suggestion of DCRNN Li et al. (2017).

Table 5: Long term global station weather forecasting performance comparison. The best results are highlighted in **bold**, while the second-best results are underlined.

| Baselines | Lead Time | Temperature | | Dewpoint | | Wind Rate | | Wind Direc. | | Sea Level | |
|---|---|---|---|---|---|---|---|---|---|---|---|
| | | MAE | MSE | MAE | MSE | MAE | MSE | MAE | MSE | MAE | MSE |
| NWP Model | 24 | 1.76 | 7.39 | 1.85 | 7.94 | 1.48 | 4.53 | 63.8 | 7158.3 | **0.86** | **2.68** |
| | 72 | **1.87** | **8.01** | **1.94** | **8.48** | 1.52 | 4.76 | 72.4 | 8215.6 | **1.06** | **3.31** |
| | 120 | **1.99** | **8.79** | **2.14** | **10.87** | 1.58 | **5.11** | 75.4 | 8647.7 | **1.38** | **5.15** |
| | 168 | **2.15** | **10.06** | **2.32** | **12.56** | 1.66 | 5.59 | 78.3 | 8945.7 | **1.87** | **9.52** |
| Pyraformer | 24 | 1.75 | 6.92 | 1.83 | 7.88 | 1.30 | 3.58 | 61.8 | 6930.2 | 1.90 | 9.72 |
| | 72 | 2.47 | 13.03 | 2.67 | 15.39 | 1.52 | 4.97 | 72.0 | 8222.4 | 3.76 | 33.67 |
| | 120 | 2.77 | 16.04 | 3.00 | 18.95 | 1.59 | 5.37 | 75.1 | 8610.7 | 4.43 | 43.91 |
| | 168 | 2.95 | 17.95 | 3.20 | 21.06 | 1.61 | 5.56 | 76.4 | **8773.5** | 4.77 | 49.97 |
| S$^2$Transformer | 24 | **1.47** | **4.99** | **1.52** | **5.67** | **1.17** | **3.09** | **55.5** | **6135.7** | 1.26 | 4.08 |
| | 72 | 2.20 | 10.32 | 2.36 | 12.33 | **1.46** | **4.62** | **68.9** | **7934.7** | 3.37 | 27.54 |
| | 120 | 2.65 | 14.54 | 2.87 | 17.48 | **1.54** | 5.13 | **73.7** | **8601.2** | 4.42 | 44.56 |
| | 168 | 2.87 | 16.88 | 3.12 | 20.20 | **1.59** | **5.42** | **75.7** | 8882.5 | 4.92 | 53.55 |

# E More Performance Comparison

## E.1 Long Term Forecasting Result

As shown in Table 5, with the increase of forecasting lead time, the error of time series forecasting methods gradually increases: except for wind speed and wind direction, their performance on almost all variables is inferior to that of physics-based numerical weather prediction models. In contrast, NWP models produce more stable predictions—partly because they are typically trained on a larger scale and more abundant data, allowing them to deliver robust global atmospheric forecasts; partly because an increase in forecast duration amplifies the nonlinear dynamics of the atmospheric system, which pure data-driven methods struggle to fully capture. In the future, we intend to explore the emerging paradigm that integrates data-driven and physics-informed approaches to enhance the model's capability of predicting long-term weather processes.

## E.2 Comparison with Physics-informed Models

To ensure a fair and comprehensive evaluation, we conduct additional experiments comparing S$^2$Transformer with two state-of-the-art physics-informed baselines, AirPhyNet (Hettige et al., 2024) and Air-DualODE (Tian et al., 2025), on the WEATHER-5K dataset. These models are specifically engineered to integrate physical constraints or differential equations into neural architectures for environmental forecasting. As summarized in Table 6, the key observations are as follows:

S$^2$Transformer achieves the lowest MAE and MSE across all core meteorological variables compared to the two physics-informed baselines. Notably, physics-informed models like Air-DualODE rely on intricate neural ODE modules for enforcing physical constraints, leading to higher memory usage and slower inference. In stark contrast, S$^2$Transformer maintains a lightweight, highly scalable architecture while outperforming these counterparts, underscoring its unique competitive edge.

Table 6: Performance comparison with physics-informed models on the WEATHER-5K dataset. The best results are highlighted in **bold**.

| Variable | Temperature | | Dewpoint | | Wind Rate | | Wind Direc. | | Sea Level | |
|---|---|---|---|---|---|---|---|---|---|---|
| Metric | MAE | MSE | MAE | MSE | MAE | MSE | MAE | MSE | MAE | MSE |
| AirPhyNet | 3.71 | 25.99 | 2.49 | 13.75 | 1.50 | 4.70 | 67.87 | 7660.30 | 2.48 | 14.62 |
| Air-DualODE | 3.20 | 19.35 | 2.12 | 10.27 | 1.49 | 4.56 | 67.26 | 7640.32 | 2.46 | 14.04 |
| S$^2$Transformer | **1.47** | **4.99** | **1.52** | **5.67** | **1.17** | **3.09** | **55.50** | **6135.69** | **1.26** | **4.08** |

Table 7: Sensitivity analysis of subgraph granularity on the WEATHER-5K dataset. Performance metrics are reported for all nodes and boundary nodes (defined under the default 64-subgraph partition). BS and Mem denote the training batch size and maximum GPU memory usage (GB), respectively.

| Subgraphs | All MAE | All MSE | Boundary MAE | Boundary MSE | BS | Mem |
|---|---|---|---|---|---|---|
| 16 | 12.13 | 1224.86 | 12.22 | 1264.10 | 64 | 61.5 |
| 64 | 12.19 | 1230.70 | 12.28 | 1270.54 | 64 | 52.0 |
| 128 | 12.25 | 1240.42 | 12.42 | 1291.76 | 64 | 46.2 |

## F  More Analysis of Graph Partitioning Strategy

To further investigate the robustness and parameter sensitivity of our graph partitioning strategy, this section provides an in-depth discussion and empirical analysis regarding the selection of subgraph granularity and the influence of partitioning stochasticity.

**Impact of the Number of Subgraphs.** Our default configuration of 64 subgraphs is selected to balance predictive performance with computational efficiency optimally. Utilizing fewer subgraphs (e.g., 16 or 32) ensures that most adjacent nodes reside within the same partition, allowing the model to capture comprehensive local correlation patterns and enhance accuracy. However, this approach incurs significant computational redundancy, as excessively large subgraphs necessitate calculations across overly extended spatial scopes. Conversely, increasing the subgraph count (e.g., 96) reduces the nodes per partition but sharply increases the number of boundary stations. For these stations, spatial correlations with neighbors in different subgraphs are truncated, leading to elevated prediction errors.

Targeted experiments, as shown in the Table 7, further analyze the impact of subgraph granularity on efficiency and forecasting performance. As the number of subgraphs increases, MAE and MSE for both all nodes and boundary nodes exhibit a clear upward trend, confirming that finer partitioning exacerbates spatial consistency loss. While 16 subgraphs yield the lowest MAE, they require 22% more memory. The 64-subgraph setting strikes a near-optimal balance, reducing memory consumption by 15.4% while sacrificing only 0.5% in all-node MAE compared to the 16-subgraph version.

Reducing the number of subgraphs to one—equivalent to no partitioning—does not represent a performance upper bound. In this configuration, the model reduces to a global attention architecture that lacks geographic inductive bias. This forces the model to capture all pairwise correlations, which introduces spurious noise from distant nodes and impairs generalization. Furthermore, the quadratic complexity of full-graph attention becomes computationally prohibitive. Thus, proper partitioning acts as a necessary regularizer that suppresses spatial noise while ensuring scalability. While our current fixed-parameter approach is effective, exploring adaptive subgraph selection remains a promising direction for future work.

**Sensitivity to Partition Randomness.** Since the METIS algorithm employs randomized seeds for initial label assignment and perturbation optimization, we evaluate the model's sensitivity to these variations. We conducted experiments using three distinct random seeds under the same partitioning constraint ($P = 64$)

Table 8: Model $S^2$Transformer performance across different METIS partitioning seeds on the NCEI Global dataset.

| NCEI Global | Wind | | Temp | |
|---|---|---|---|---|
| Metric | MAE | MSE | MAE | MSE |
| Seed=15 | 1.304 | 3.616 | 1.884 | 7.524 |
| Seed=42 | 1.307 | 3.631 | 1.880 | 7.516 |
| Seed=2025 | 1.305 | 3.620 | 1.874 | 7.509 |

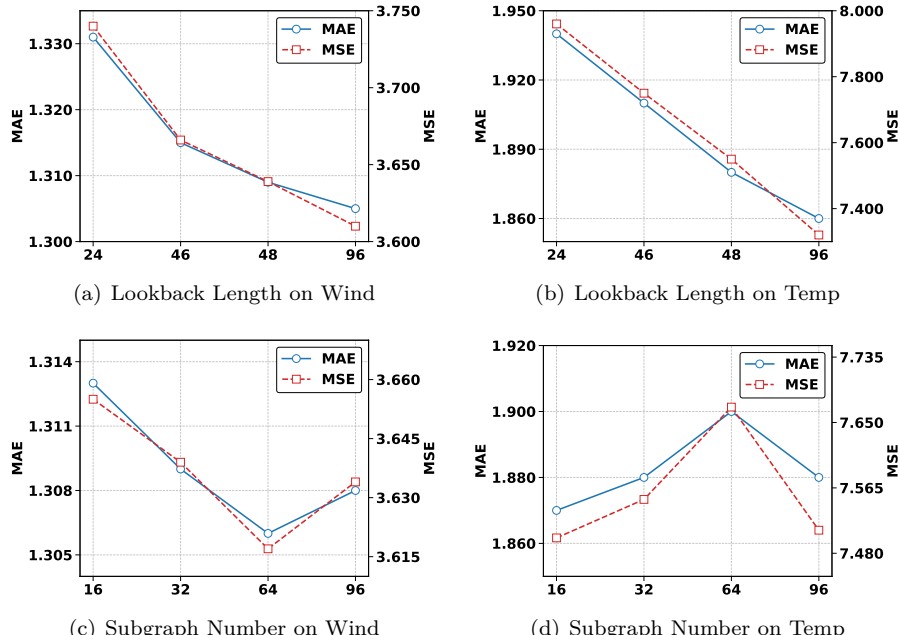

(a) Lookback Length on Wind      (b) Lookback Length on Temp

(c) Subgraph Number on Wind      (d) Subgraph Number on Temp

Figure 5: More Parameter sensitivity analysis

on the NCEI Global dataset. As summarized in Table 8, the performance variations across different seeds are negligible. This consistency confirms that the $S^2$Transformer is highly robust to the stochastic nature of the graph partitioning process.

# G  More Parameters Sensitivity Analysis

We evaluate the impact of lookback window length $T$ and initial subgraph number $P$ on model performance on the NCEI Global dataset. As shown in Figure 5, increasing $T$ improves performance for both datasets. However, longer input sequences cause rapid surges in computational and memory costs for spatiotemporal GNNs, leading most existing models to rely on short-term historical windows, severely limiting their performance (Han et al., 2024a). In contrast, our model achieves spatial scalability while accommodating larger lookback windows to boost performance. Next, we evaluate the impact of the initial subgraph number $P$ (ranging from 16, 32, 64 to 96). A larger $P$ means fewer nodes per subgraph, enabling more precise modeling of local spatial correlations; conversely, a smaller $P$ increases nodes per subgraph, expanding the local spatial scope but raising memory and computational costs. Experimental results show that despite the same number of stations, the performance trends and optimal $P$ values vary across datasets with different variables. This is attributed to complex interactions between local and global influences, indicating limitations in treating $P$ merely as a hyperparameter without adjusting it to balance local and global effects. Exploring adaptive selection of $P$ will be part of our future work.

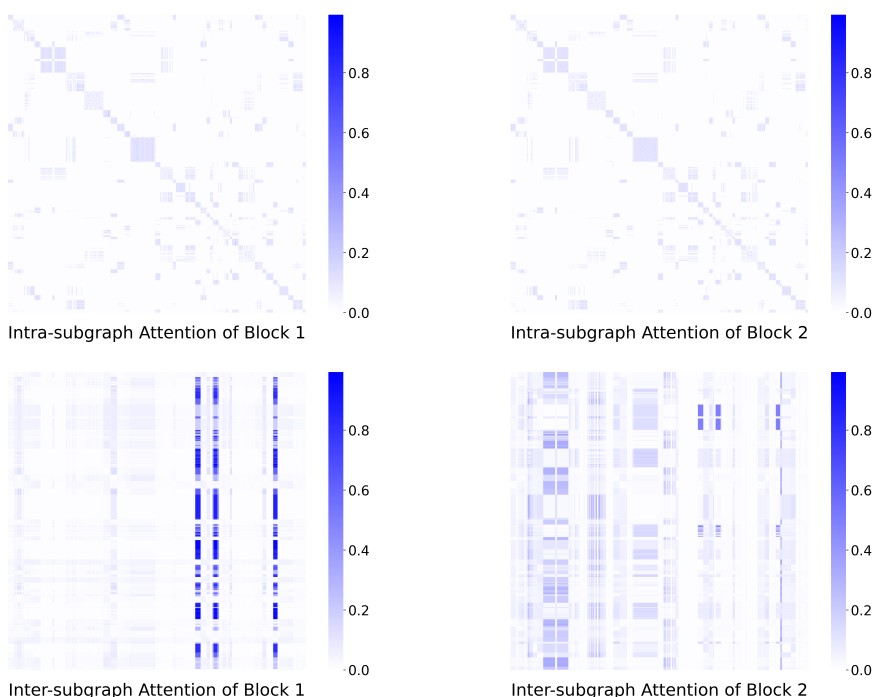

Figure 6: Visualization of intra-subgraph and inter-subgraph attention matrices for S$^2$Transformer.

## H  More Visualization

**Attention matrix.**  We further visualize the intra-subgraph and inter-subgraph attention matrices of S$^2$Transformer.  As shown in Figure 6, the two-layer attention mechanism effectively separates local and global information (i.e., structured spatial correlation modeling): in local modeling, the model filters out global noise (the attention matrix has non-zero values only locally, showing a sparse high-rank pattern); while global modeling is achieved through subgraph aggregation and learning attention between subgraphs (the attention matrix exhibits high weights and a sparse low-rank pattern). We propose that this low-rank pattern can be interpreted as the discovery of key hubs in large-scale spatiotemporal networks.

**Global Station Forecasting result.**  As shown in Figure 7, we plot the prediction errors of different models for the temperature variable in the WEATHER-5K Dataset from a global perspective (brighter colors indicate larger prediction errors at the corresponding stations).  It can be observed that all models show an upward trend in prediction errors over time; among them, S$^2$Transformer achieves higher accuracy in temperature prediction for high-latitude stations, while baseline models exhibit significantly larger prediction errors at these stations.  This advantage originates from two designs of the proposed model: first, spherical harmonic positional encoding endows it with a clear definition worldwide (including polar regions), enabling better distinction of meteorological stations distributed globally; second, the attention mechanism with progressively expanded receptive fields conforms to physical diffusion laws.

**Single Station Forecasting result.**  As shown in Figure 8 and Figure 9, we plot the predictions of different models for the temperature variable in the NCEI Global Dataset from a single-station perspective.  It can be observed that S$^2$Transformer outperforms other baseline models in the modeling of seasonal, peak, and stationary sequence values.  This is attributed to the model's ability to effectively capture local spatial correlations and leverage changes in neighboring nodes to achieve more accurate forecasting.

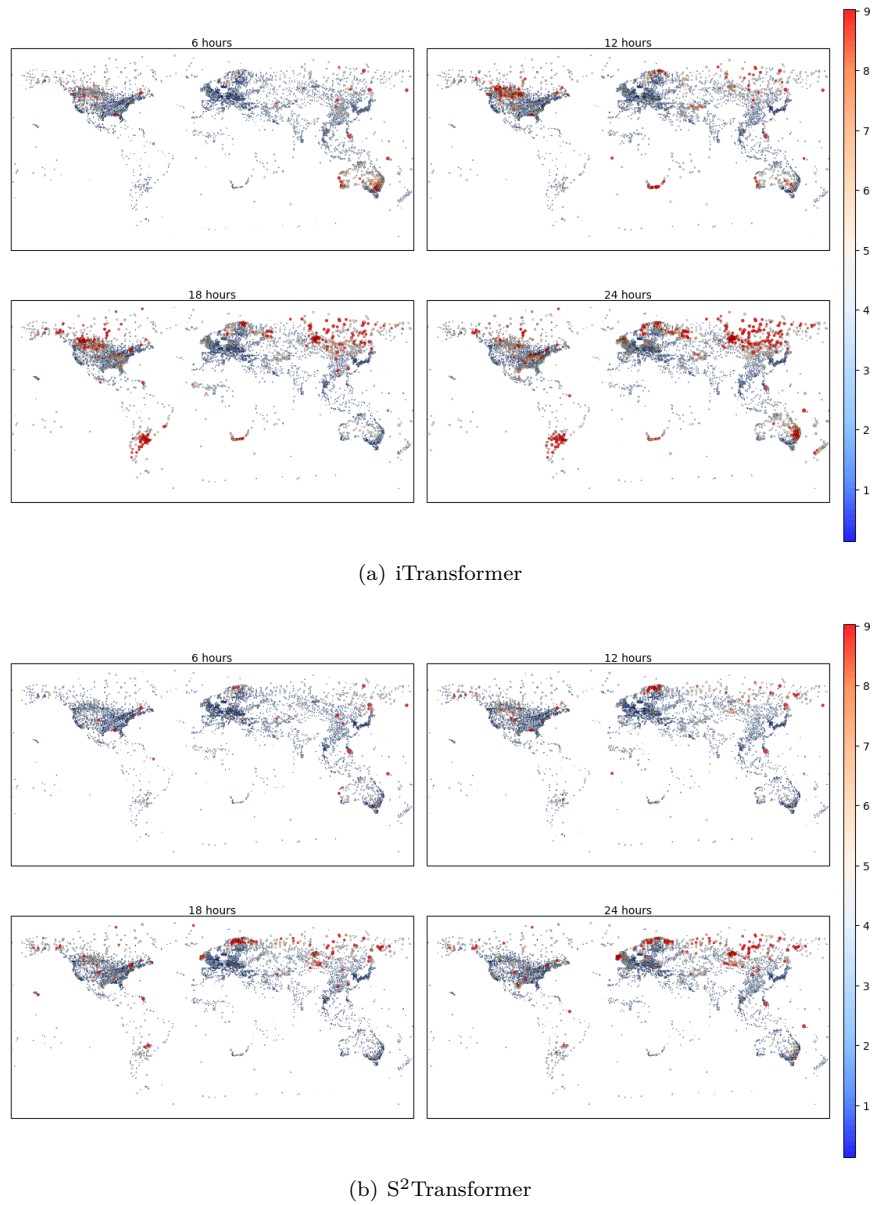

(a) iTransformer

(b) S$^2$Transformer

Figure 7: Comparison of global station forecasting errors.

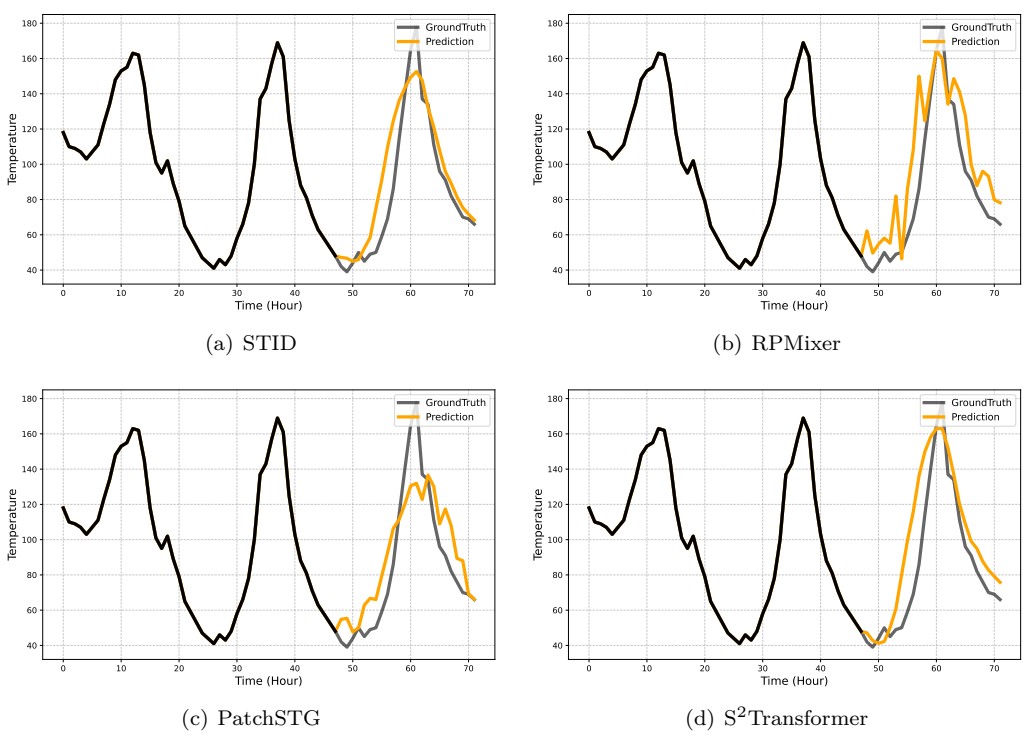

Figure 8: Comparison of single station forecasting results (seasonal sequences).

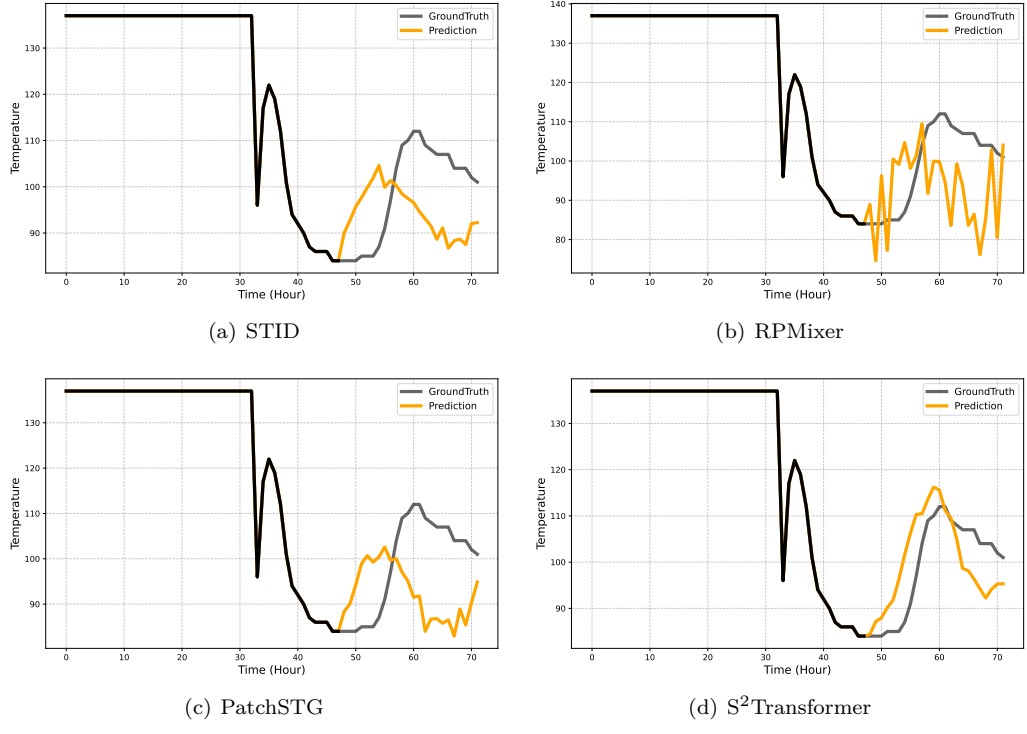

Figure 9: Comparison of single station forecasting results (peak sequences).

