# OpenReview forum: "S$^2$Transformer: Scalable Structured Transformers for Global Station Weather Forecasting"
_TMLR — Accepted by TMLR_

### Review · Reviewer_ztAG · 2025-10-14

**Summary Of Contributions:**

This paper modifies the Transformer architecture to improve global station weather forecasting, taking into account spatial relationships between different stations. The authors first construct a spatial graph where two stations are connected if they are close enough. After partitioning the graph, they introduce a spatial structured attention (SSA) block, which first applies attention locally (intra-subgraph), then globally (inter-subgraph). The authors further adapt the positional encodings and add a spatial attention bias. They also propose a multiscale architecture across layers, moving from finer-grained to coarser subgraphs. The proposed approach is evaluated on 2 short-term weather forecasting datasets (WEATHER-5K and NCEI Global), often outperforming existing baselines.

**Audience:**

Yes

**Audience Explanation:**

The paper proposes changes to the Transformer architecture to better capture spatial relationships, which should be interesting to some of the TMLR audience.

**Claims And Evidence:**

Yes

**Claims Explanation:**

Mostly yes. The proposed approach is compared to a variety of reasonable baselines and often outperforms them. One limitation of the evaluation is that the time frame is fixed (48-hour history, 24-hour prediction).

The authors include a complexity analysis of their method, as well as memory usage and training time statistics.

The authors conduct an ablation study for most of the components of their proposed architecture, although additional evidence to demonstrate the impact of using different scales across layers (section 4.3) would be welcome.

**Requested Changes:**

**Critical**

- Add an ablation study for graph partition size and the impact of using different scales across layers.

**Would strengthen work**

- Could the approach be extended to take into account microclimates, where nearby stations might have fairly different weather?

- On line 318/319, is the positional encoding completely removed or just simplified?

- In addition to average results, can you include the standard deviation?

- On line 271/272, did you try the opposite approach (single vs joint predictions) on the 2 datasets?

- On line 336, you mention that increasing the number of blocks leads to worse results. Could this be mitigated with better regularization?

- The input and output linear transformations appear to be potential bottlenecks to handle longer sequences. Could you discuss how you could extend the approach to better support longer timeframes?

---

> ### Author Response · Authors · 2025-12-14
> **Response to Reviewer ztAG**
>
> **Q1: One limitation of the evaluation is that the time frame is fixed (48-hour history, 24-hour prediction).**
>
> In our submitted paper, Appendix E analyzes the model’s performance with varying history lengths (e.g., 24, 48, 96 hours) on the NCEI Global dataset, while Appendix D presents results for extended prediction  lengths (e.g., 72, 120, 168 hours) across the WEATHER-5K dataset.
>
> **Q2: Add an ablation study for graph partition size and the impact of using different scales across layers.**
>
> The ablation study for graph partition size is already presented in Appendix E of our submission, where we evaluate model performance across varying partition sizes (e.g., 16, 32, 64, 96 subgraphs). We further conducted experiments to analyze how graph partition size influences efficiency and performance. (See our response to Reviewer **mVgX's Q1** for details.)
>
> However, we acknowledge the omission of an ablation study on cross-layer multi-scale impacts. To address this, we conducted additional experiments, with the results presented below:
>
> | MAE(MSE)         |   Temperature   |    Dewpoint     |    Wind Rate    |     Wind Direc.     |    Sea Level    |
> | :--------------- | :-------------: | :-------------: | :-------------: | :-----------------: | :-------------: |
> | w/o. MA          |   1.59 (5.12)   |   1.57 (5.81)   |   1.19 (3.10)   |   55.86 (6188.84)   |   1.28 (4.18)   |
> | S$^2$Transformer | **1.47 (4.99)** | **1.52 (5.67)** | **1.17 (3.09)** | **55.50 (6135.69)** | **1.26 (4.08)** |
>
>  "w/o. MA" denotes the model without the hierarchical multi-scale architecture (using single-scale blocks only). These experiments demonstrate that our hierarchical multi-scale strategy outperforms single-scale configurations, confirming the value of structured scale progression in capturing multi-level spatial correlations. These findings will be added to the revised manuscript. Thank you again for this valuable suggestion.
>
> **Q3: Could the approach be extended to take into account microclimates, where nearby stations might have fairly different weather?**
>
> We appreciate the reviewer’s insightful question, as microclimates (e.g., urban heat islands, slope-related temperature differences) indeed cause notable weather discrepancies between geographically adjacent stations—a practical challenge for GSWF that deserves attention.
>
> Our current S²Transformer already has partial adaptability to such scenarios. First, the **Intra-subgraph Attention** enables fine-grained modeling of local correlations, avoiding overgeneralizing "spatial proximity" to dissimilar stations. Second, while the **Spatial Attention Bias** currently adjusts weights via shortest path distance, it can be extended to incorporate microclimate features (e.g., topography, vegetation type) as additional biases—allowing the model to automatically identify cases where adjacent stations have large climatic differences. To fully address this, we propose three targeted extensions for future work:
>
> 1. Incorporate microclimate-related features into input embeddings to capture local environmental influences upfront.
> 2. Modify attention weight calculation by adding a "microclimate similarity factor" to reduce correlation weights between adjacent but climatically dissimilar stations.
> 3. Optimize subgraph partitioning by integrating microclimate types, ensuring subgraphs contain stations with consistent local climate patterns for more accurate correlation modeling.
>
> **Q4: On line 318/319, is the positional encoding completely removed or just simplified?**
>
> We confirm the spherical harmonic positional encoding is fully removed in "w/o. SH" (Line 318/319)—--no simplification or alternative is used. Unlike the full model (Eq. 3, SH fused with time-series embeddings), "w/o. SH" only uses linear-transformed raw observations. The higher MAE/MSE in Table 4 confirms SH’s irreplaceable spatial value, and this strict removal ensures valid contribution verification.
>
> **Q5: On line 271/272, did you try the opposite approach (single vs joint predictions) on the 2 datasets?**
>
> For WEATHER-5K, we use a unified architecture for joint variable prediction, consistent with previous works [1]. For NCEI Global, we train and test separately for each variable, aligning with previous works [2]. This alignment ensures our results are directly comparable to existing baselines, without biases from inconsistent prediction paradigms.
>
> [1] How far are today’s time-series models from real-world weather forecasting applications? arXiv preprint arXiv:2406.14399,2024.
>
> [2] Interpretable weather forecasting for worldwide stations with a unified deep model. Nature Machine Intelligence, 2023.

---

> ### Author Response · Authors · 2025-12-14
> **Response to Reviewer ztAG cont.**
>
> **Q6: In addition to average results, can you include the standard deviation?**
>
> We agree to include standard deviation alongside average results. Due to space limitation, we present key results (mean ± std) from 3 repeated experiments on the NCEI Global dataset below:
>
> | NCEI Global      | Wind MAE            | Wind MSE            | Temp MAE            | Temp MSE            |
> | ---------------- | ------------------- | ------------------- | ------------------- | ------------------- |
> | Corrformer       | 1.320$\pm$0.028     | 3.925$\pm$0.039     | 1.910$\pm$0.024     | 7.840$\pm$0.128     |
> | iTransformer     | 1.332$\pm$0.004     | 3.890$\pm$0.022     | 1.905$\pm$0.018     | 7.551$\pm$0.051     |
> | S$^2$Transformer | **1.303$\pm$0.005** | **3.630$\pm$0.020** | **1.874$\pm$0.008** | **7.535$\pm$0.032** |
>
> S$^2$Transformer achieves the lowest MAE and MSE across all metrics with standard deviations comparable to or smaller than baselines, which indicates both superior prediction accuracy and stable performance. The small standard deviations further confirm the reliability of our model’s results.
>
> **Q7: On line 336, you mention that increasing the number of blocks leads to worse results. Could this be mitigated with better regularization?**
>
> The performance drop with more blocks ($L$) mainly comes from overfitting: extra blocks expand model capacity, leading to over-adaptation to training data. Theoretically, better regularization could mitigate this by constraining excessive parameter learning. However, our experiments already used standard regularization (e.g., 0.1 dropout) but we did not observe corresponding performance gains when increasing the number of blocks. In future work, we will further explore more advanced regularization strategies (e.g., adaptive dropout, weight decay tuning) to test if they can alleviate this performance drop.
>
> **Q8: The input and output linear transformations appear to be potential bottlenecks to handle longer sequences. Could you discuss how you could extend the approach to better support longer timeframes?**
>
> We appreciate the reviewer’s insight that input/output linear transformations may bottleneck longer sequences. To better support extended timeframes, we propose two actionable improvements:
>
> 1. **Integrate temporal modeling modules**: Pair linear transformations with specialized temporal blocks (e.g., causal CNNs for local temporal patterns or SparseTSF [3] for long-range temporal mixing) to enhance feature extraction from long sequences.
> 2. **Adopt a hybrid prediction paradigm**: For ultra-long timeframes, use linear layers for short-term feature mapping, and complement them with physics-informed constraints (aligning with atmospheric dynamics) to ensure stable long-term predictions.
>
> [3] SparseTSF: modeling long-term time series forecasting with 1k parameters. International Conference on Machine Learning, 2024.

---

### Review · Reviewer_mVgX · 2025-10-15

**Summary Of Contributions:**

The paper proposes a novel transformer-based framework, S²Transformer, for global station weather forecasting. The approach introduces a graph partition mechanism and a multi-scale spatial structure, enabling the model to capture long-range global correlations while preserving scalability. Experimental results demonstrate both the effectiveness and computational efficiency of the proposed method.

**Audience:**

Yes

**Audience Explanation:**

The paper uses a new transformers framework to deal with GSFW, which could be considered as the application of graph representation learning.

**Claims And Evidence:**

Yes

**Claims Explanation:**

The authors make three primary claims:
• Global correlations are essential for global station weather forecasting (GSWF) tasks;
• The proposed S²Transformer is more scalable to large graphs compared to existing transformer-based architectures; and
• S²Transformer achieves performance that is either superior to or on par with other state-of-the-art GSWF methods.

These claims are well-supported by the presented experimental results, which include comprehensive evaluations demonstrating both scalability and predictive accuracy across multiple datasets.

**Requested Changes:**

• The authors should provide a more detailed discussion regarding the choice of the number of subgraphs in the initial partition. Currently, only the changes in MAE and MSE for two tasks are reported under different subgraph counts. It would be beneficial to include further analysis on how varying the number of subgraphs affects model performance and spatial consistency, particularly for boundary stations located between subgraphs.

• Since both the authors’ claims and experimental results emphasize the importance of the graph partition strategy to overall performance, it would strengthen the paper to include additional analysis on the sensitivity of the model to partition randomness. For example, do different random seeds used by METIS (under the same partitioning constraints) lead to noticeable performance variations?

• As GSWF can be formulated as a node-level forecasting task, and the main contribution of S²Transformer lies in its scalability, it would be informative to compare it against existing linear transformer architectures, such as NodeFormer [1], which can handle graphs with over 100K nodes. Including such comparisons would help better position S²Transformer within the landscape of efficient transformer-based models.

Reference:
[1] Wu Q, Zhao W, Li Z, Wipf DP, Yan J. Nodeformer: A scalable graph structure learning transformer for node classification. Advances in Neural Information Processing Systems. 2022 Dec 6;35:27387-401.

---

> ### Author Response · Authors · 2025-12-14
> **Response to Reviewer mVgX**
>
> **Q1: The authors should provide a more detailed discussion regarding the choice of the number of subgraphs in the initial partition. It would be beneficial to include further analysis on how varying the number of subgraphs affects model performance and spatial consistency, particularly for boundary stations located between subgraphs.**
>
> We appreciate the reviewer’s suggestion to elaborate on the selection of the initial number of subgraphs and its impacts. Our default setting of 64 subgraphs is determined to optimally balance model performance and computational efficiency.
>
> When using fewer subgraphs (e.g., 16 or 32), most nodes and their adjacent counterparts fall within the same subgraph, this enables the model to capture more comprehensive correlation patterns between neighboring nodes, which is beneficial for improving prediction accuracy. However, it also introduces redundant computation, as excessively large subgraphs require unnecessary calculations across overly extended spatial scopes. In contrast, more subgraphs (e.g., 96) reduce the number of nodes per subgraph while significantly increasing the count of boundary stations. For these boundary stations, spatial correlations with adjacent nodes that belong to other subgraphs cannot be fully modeled; this disruption of spatial associations ultimately leads to elevated prediction errors.
>
> We also conducted targeted experiments to analyze how the number of subgraphs influences model efficiency and forecasting performance, with key results presented in the table below.
> | Subgraph Number | All nodes MAE | All nodes MSE | Boundary nodes MAE | Boundary nodes MSE | Training BS | Max Mem |
> | --------------- | ------------- | ------------- | ------------------ | ------------------ | ----------- | ------- |
> | 16              | 12.13         | 1224.86       | 12.22              | 1264.10            | 64          | 61.5    |
> | 64              | 12.19         | 1230.70       | 12.28              | 1270.54            | 64          | 52.0    |
> | 128             | 12.25         | 1240.42       | 12.42              | 1291.76            | 64          | 46.2    |
>
> Note: Boundary node metrics are calculated by first defining "boundary stations" under the 64-subgraph partition (our default) and then tracking their performance across different subgraph number settings. The remaining metrics in the table refer to the training batch size (Training BS), maximum memory consumed during training (Max Mem, in GB).
>
> **Experimental Result Analysis**
>
> - As the number of subgraphs increases, MAE/MSE for both all nodes and boundary nodes shows a clear upward trend, which confirms that more subgraphs exacerbate spatial consistency loss for boundary stations. In contrast, 16 subgraphs yield the lowest MAE but at the cost of 22% higher memory usage.
> - The 64-subgraph setting achieves a near-optimal balance: while its all-node MAE is only 0.5% higher than that of 16 subgraphs, it reduces memory consumption by 15.4%.
>
> **Q2:  It would strengthen the paper to include additional analysis on the sensitivity of the model to partition randomness. For example, do different random seeds used by METIS (under the same partitioning constraints) lead to noticeable performance variations?**
>
> We agree with the reviewer that analyzing sensitivity to partition randomness is critical for validating our graph partition strategy. This is because METIS-based graph partitioning randomly assigns initial subgraph labels to nodes and uses seed-controlled random perturbations during optimization to avoid local optima, leading to slight differences in subgraph composition even under the same constraints. To address this, we conducted additional experiments: under the same constraint (64 subgraphs, consistent with our default setting), we tested S²Transformer with 3 different random seeds for METIS partitioning on the NCEI Global dataset:
>
> | NCEI Global | Wind MAE | Wind MSE | Temp MAE | Temp MSE |
> | ----------- | -------- | -------- | -------- | -------- |
> | seed=15     | 1.304    | 3.616    | 1.884    | 7.524    |
> | seed=42     | 1.307    | 3.631    | 1.880    | 7.516    |
> | seed=2025   | 1.305    | 3.620    | 1.874    | 7.509    |
>
> The results show minimal performance variations across seeds for core meteorological variables. This confirms the model’s strong robustness to randomness in METIS partitioning.

---

> ### Author Response · Authors · 2025-12-14
> **Response to Reviewer mVgX cont.**
>
> **Q3: It would be informative to compare it against existing linear transformer architectures, such as Nodeformer [1], which can handle graphs with over 100K nodes. Including such comparisons would help better position S²Transformer within the landscape of efficient transformer-based models.**
>
> We appreciate the reviewer’s valuable suggestion. To address this, we conducted additional experiments comparing S²Transformer and Nodeformer on the WEATHER-5K dataset, with key results presented below:
>
> |                  | Temperature | Dewpoint | Wind Rate | Wind Direc. | Sea Level | Training BS | Max Mem  |
> | ---------------- | :---------: | -------- | --------- | ----------- | --------- | ----------- | -------- |
> | Nodeformer       |    1.74     | 1.80     | 1.27      | 60.6        | 1.69      | 16          | 133.9    |
> | S$^2$Transformer |  **1.47**   | **1.52** | **1.17**  | **55.5**    | **1.26**  | **64**      | **52.0** |
>
> The first five metrics are measured in MAE, while the remaining ones refer to training batch size (Training BS), maximum training memory (Max Mem, in GB).
>
> **Experimental Result Analysis**
>
> - Nodeformer underperforms S²Transformer across all core meteorological variables, which stems from its original design for static node classification—--a task without the need to model temporal dynamics or domain-specific spatial correlations.
> - While Nodeformer achieves $\mathcal{O}(N)$ linear complexity via Gumbel sampling, it incurs prohibitive memory overhead due to storing intermediate results for multi-step sampling. The high memory usage forces Nodeformer to use a small batch size, slowing convergence and limiting scalability.
>
> In summary, while Nodeformer excels at large-scale static node classification, S²Transformer’s performance and efficiency advantages are tailored to the unique demands of GSWF, confirming its value in the landscape of efficient transformer-based models for weather forecasting.
>
> [1] Nodeformer: A scalable graph structure learning transformer for node classification. Neural Information Processing Systems, 2022.

---

> > ### Comment · Reviewer_mVgX · 2026-01-05
> > **Thank you for the response**
> >
> > I have some further questions:
> > Q1: Is there a performance upper bound for fewer subgraphs? Would the model get the best performance when the number of subgraphs is 1?
> >
> > Q3: I am somewhat confused about the batch size reported for Nodeformer. Based on my understanding, the effective training batch size of Nodeformer should be 1, unless multiple historical time windows are constructed as independent samples for feature forecasting. My intention is not to ask the authors to directly compare their method with Nodeformer. Rather, based on the results in Q1, we observe that the performance of S2 improves as the number of subgraphs decreases. This suggests that reducing the number of subgraphs is beneficial. Therefore, I encourage the authors to consider incorporating linear Transformer techniques into their method, which could potentially reduce the required number of subgraphs while maintaining or improving performance.

---

> > > ### Author Response · Authors · 2026-01-06
> > > **Response to Reviewer mVgX**
> > >
> > > **Q1**: **Is there a performance upper bound for fewer subgraphs? Would the model get the best performance when the number of subgraphs is 1?**
> > >
> > > When the number of subgraphs is set to 1 (i.e., no graph partitioning), our model essentially reduces to iTransformer. As shown in the comparative experiments in our submitted paper, this setting fails to deliver optimal performance:
> > >
> > > - **Degraded generalization**: For global meteorological systems, geographically adjacent stations typically exhibit similar local climate patterns. Without subgraph-induced geographic bias, the model must model all pairwise correlations across the entire graph, easily introducing spurious correlations as noise on large-scale datasets and thus impairing generalization **(see our response to Reviewer U1EA’s Q4 for detailed evidence)**.
> > > - **Prohibitive overhead**: Full-graph attention entails unnecessary computations over an overly broad spatial scope, drastically increasing memory usage and training time.
> > >
> > > Proper subgraph partitioning injects geographic bias to boost generalization and reduce computational and memory overhead, while overly fine-grained partitioning sharply increases boundary station counts. For these stations, cross-subgraph spatial correlations cannot be fully modeled, and this disruption of spatial associations ultimately leads to elevated prediction errors.
> > >
> > > Thus, performance plateaus (or even degrades) as the number of subgraphs decreases, and the single-subgraph setting fails to yield optimal performance. Our default 64-subgraph configuration is chosen to balance model performance and computational efficiency optimally. Naturally, this fixed-parameter approach has its limitations, and exploring adaptive subgraph number selection will be part of our future work.
> > >
> > > ---
> > >
> > > **Q2:  I encourage the authors to consider incorporating linear Transformer techniques into their method, which could potentially reduce the required number of subgraphs while maintaining or improving performance.**
> > >
> > > We appreciate the reviewer’s valuable suggestion. As noted in our response to Q1, moderately reducing the number of subgraphs (while avoiding overly few subgraphs) indeed helps boost model performance, but this comes with a marked increase in attention computation overhead. Linear Transformers (e.g., Nodeformer) perfectly resolve this dilemma—they hold the potential to cut down subgraph counts while keeping computational costs in check. Our current research focuses on the synergistic design of subgraph partitioning and structured attention. Due to time limitations, we have to leave the exploration of fewer subgraphs + linear attention for our future work.

---

### Review · Reviewer_U1EA · 2025-12-08

**Summary Of Contributions:**

This paper proposes $S^2$Transformer, a scalable spatiotemporal forecasting model for global station weather prediction. Its key contribution is the introduction of a Spatial Structured Attention Block, which partitions the global spatial graph into subgraphs to jointly model local (intra-subgraph) and global (inter-subgraph) correlations through attention mechanisms. This design reduces noise and quadratic computational cost, enabling efficient long-range spatial dependency modeling. Experiments on medium- and large-scale weather station datasets show notable performance and efficiency gains over existing baselines.

**Audience:**

Yes

**Audience Explanation:**

Yes. The paper’s findings would interest part of TMLR’s audience, especially researchers in spatiotemporal forecasting, graph-based transformers, and AI for earth system modeling. While not groundbreaking, its structured attention design and scalable architecture address a relevant and timely problem in large-scale weather prediction.

**Broader Impact Concerns:**

No major ethical concerns identified. The model focuses on weather forecasting and does not involve sensitive or private data. A short Broader Impact Statement discussing responsible deployment in climate and disaster prediction would be sufficient.

**Claims And Evidence:**

Yes

**Claims Explanation:**

The paper presents quantitative results on two public datasets, demonstrating that $S^2$Transformer achieves significant advantages in MAE/MSE metrics, memory consumption, and inference speed. It also conducts ablation studies to validate the effectiveness of key components such as subgraph partitioning and spatial attention bias, with results consistent with the model’s design objectives.

**Requested Changes:**

1. At line 318, the model’s core mechanism (Intra- and Inter-subgraph Attention) relies entirely on the quality of the subgraph partitioning. However, the authors only compare two variants (removing METIS and using random partitioning), without discussing the impact of partitioning strategy, robustness, or potential bias.
2. Table 3 includes only general spatiotemporal baselines such as MTGNN, iTransformer, and Corrformer, but omits hybrid physics-driven or physics-informed models, which are essential for a fair and comprehensive comparison.
3. The paper does not provide a formal mathematical definition of “structured spatial correlation.” I recommend that the authors clarify this concept, either by including a precise formulation in the appendix or by adding comparative experiments to better illustrate what “structured” represents in this context.

---

> ### Author Response · Authors · 2025-12-14
> **Response to Reviewer U1EA**
>
> **Q1: At line 318, the model’s core mechanism (Intra- and Inter-subgraph Attention) relies entirely on the quality of the subgraph partitioning. However, the authors only compare two variants (removing METIS and using random partitioning), without discussing the impact of partitioning strategy, robustness, or potential bias.**
>
> We sincerely appreciate the reviewer’s incisive observation on the critical role of subgraph partitioning for our model’s core Intra- and Inter-subgraph Attention mechanism. **Reviewer mvgX** also raises the similar concerns, which have been addressed in our prior responses, and we kindly request you to refer to them listed as follows.
>
> 1. For the impact of the partitioning strategy, a detailed discussion of subgraph number selection and its trade-offs among performance, computational efficiency, and boundary station spatial consistency is provided **in our response to Reviewer mVgX’s Q1.**
> 2. Regarding partitioning robustness, we validated the model’s insensitivity to METIS partitioning randomness via multi-seed experiments, with negligible performance fluctuations demonstrated **in our response to Reviewer mVgX’s Q2.**
> 3. On potential partitioning bias, we clarify that our strategy is tailored to global station weather forecasting. For global meteorological systems, geographically adjacent stations typically share similar local climate patterns, aligning with Tobler’s first law of geography. Thus, the spatial bias we introduce based on geographic proximity in partitioning is a reasonable prior that enhances (rather than distorts) the model’s ability to capture local spatial correlations.
>
>
> **Q2: Table 3 includes only general spatiotemporal baselines such as MTGNN, iTransformer, and Corrformer, but omits hybrid physics-driven or physics-informed models, which are essential for a fair and comprehensive comparison.**
>
> We appreciate the reviewer’s valuable suggestion. To address this, we conducted additional experiments comparing S$^2$Transformer with two representative physics-informed baselines: AirPhyNet [1] and Air-DualODE [2] on the WEATHER-5K dataset, with key results presented below:
>
> | MAE(MSE)         |  Temperature   |    Dewpoint    |   Wind Rate    |    Wind Direc.     |   Sea Level    |
> | :--------------- | :------------: | :------------: | :------------: | :----------------: | :------------: |
> | AirPhyNet        |  3.71(25.99)   |  2.49(13.75)   |   1.50(4.70)   |   67.87(7660.30)   |  2.48(14.62)   |
> | Air-DualODE      |  3.20(19.35)   |  2.12(10.27)   |   1.49(4.56)   |   67.26(7640.32)   |  2.46(14.04)   |
> | S$^2$Transformer | **1.47(4.99)** | **1.52(5.67)** | **1.17(3.09)** | **55.50(6135.69)** | **1.26(4.08)** |
>
> S$^2$Transformer achieves the lowest MAE and MSE across all core meteorological variables compared to the two physics-informed baselines. Notably, physics-informed models like Air-DualODE rely on intricate neural ODE modules for enforcing physical constraints, leading to higher memory usage and slower inference. In stark contrast, S$^2$Transformer maintains a lightweight, highly scalable architecture while outperforming these counterparts, underscoring its unique competitive edge.
>
> [1] AirPhyNet: Harnessing Physics-Guided Neural Networks for Air Quality Prediction. International Conference on Learning Representations, 2024.
>
> [2] Air Quality Prediction with Physics-Guided Dual Neural ODEs in Open Systems. International Conference on Learning Representations, 2025.
>
> **Q3: A short Broader Impact Statement discussing responsible deployment in climate and disaster prediction would be sufficient.**
>
> We fully agree that a concise discussion of deployment in climate and disaster prediction will better highlight our work’s societal value. In the revised manuscript, we will update the Broader Impact Statement to briefly note that the S$^2$Transformer’s low-cost global station weather forecasting capability can support resource-constrained regions to access reliable weather forecasts and promote equitable access to tools for climate risk management.

---

> ### Author Response · Authors · 2025-12-14
> **Response to Reviewer U1EA cont.**
>
> **Q4: The paper does not provide a formal mathematical definition of “structured spatial correlation.” I recommend that the authors clarify this concept, either by including a precise formulation in the appendix or by adding comparative experiments to better illustrate what “structured” represents in this context.**
>
> We sincerely appreciate the reviewer’s insightful suggestion. We will address this gap in the revised version while also highlighting the existing experimental validation of the “structured” property:
>
> - A spatial correlation matrix $\boldsymbol{\Theta} \in \mathbb{R}^{N\times N}$ (where $\Theta_{ij}$ denotes the correlation strength between station $v_i$ and $v_j$) learned by a neural network is defined as **a structured spatial correlation** if it satisfies the following two constraints:
>
>   1. **Geographic proximity constraint**: For any station $v_i$, the cumulative correlation strength of its geographic $\epsilon$-neighborhood $\mathcal{N}_\epsilon(v_i)$ accounts for at least a threshold proportion $\tau$ of its total correlation strength:
>
>      $$
>      \frac{\sum_{v_j\in N_\epsilon(v_i)}\Theta_{ij}}{\sum_{v_j\in V}\Theta_{ij}} \geq \tau
>      $$
>
>   2. **Sparsity Constraint**: The sparsity of $\boldsymbol{\Theta}$ is bounded by a threshold $\rho$ to eliminate trivial noise:
>      $$
>      \frac{|\{(i,j)|\Theta_{ij}>0\}|}{N^2} \leq \rho
>      $$
>
> - The submitted manuscript has already provided solid empirical evidence for the “structured” property of the correlation learned by our model, with key results as follows:
>
>   1. The performance comparison **(Table 2, Lines 269–294)** shows that on the WEATHER-5K and NCEI Global datasets, S$^2$Transformer outperforms iTransformer (a representative unstructured attention-based method), confirming that structured correlation filters remove noise and retain critical geographic patterns.
>   2. Attention matrix visualization **(Figure 3, Lines 341–357)** compares the intra-subgraph attention of S$^2$Transformer with iTransformer’s corresponding subgraph-region attention, showing iTransformer’s attention values are scattered (max ≈0.12) across irrelevant station pairs (unstructured), while our model’s weights focus on geographically relevant nodes (structured), providing direct qualitative evidence of the learned correlation’s “structured” property.

---

### Decision · Action_Editor_11TY · 2026-02-19

**Recommendation:** Accept with minor revision

**Additional Comments:**

Several aspects require minor revision, including deeper analysis of partition robustness and sensitivity, clearer formalization of “structured spatial correlation,” more baseline comparisons, and additional ablations on partition size and architectural choices. These revisions would further strengthen the paper.

**Audience:**

Yes

**Audience Explanation:**

The paper will attract some audience in spatiotemporal learning and weather forecasting applications.

**Claims And Evidence:**

Yes

**Claims Explanation:**

This paper proposes a scalable spatiotemporal transformer that effectively captures local and global dependencies while reducing computational cost. The empirical results on different datasets are compelling and demonstrate clear efficiency gains over strong baselines. Reviewers are generally positive about this work after rebuttal.